# Forage quality declines with rising temperatures, with implications for livestock production and methane emissions

Mark A. Lee[1], Aaron P. Davis[1], Mizeck G. G. Chagunda[2], Pete Manning[3]

[1]Natural Capital and Plant Health, Royal Botanic Gardens, Kew, Richmond, TW9 3AB, UK

[2]Dairy Research and Innovation Centre, Scotland's Rural College, The Crichton, Dumfries, DG1 4TA, UK

[3]Biodiversity and Climate Research Centre, Senckenberg, Senckenberganlage 25, Frankfurt am Main, 60325, Germany

*Correspondence to:* Mark A.Lee (m.lee@kew.org)

Abstract: Livestock numbers are increasing to supply the growing demand for meat-rich diets. The sustainability of this trend has been questioned, and future environmental changes, such as climate change, may cause some regions to become less suitable for livestock. Livestock and wild herbivores are strongly dependent on the nutritional chemistry of forage plants. Nutrition is positively linked to weight gains, milk production and reproductive success, and nutrition is also a key determinant of enteric methane production. In this meta-analysis we assessed the effects of growing conditions on forage quality by compiling published measurements of grass nutritive value and combining these data with climatic, edaphic and management information. We found that forage nutritive value was reduced at higher temperatures and increased by nitrogen fertiliser addition, likely driven by a combination of changes to species identity, and changes to physiology and phenology. These relationships were combined with multiple published empirical models to estimate forage and temperature driven changes to cattle enteric methane production. This suggested a previously undescribed positive climate change feedback, where elevated temperatures reduce grass nutritive value and correspondingly may increase methane production by 0.9% with a 1 °C temperature rise and 4.5 % with a 5 °C rise (model average), thus creating an additional climate forcing effect. Future methane production increases are expected to be largest in parts of North America, Central and Eastern Europe, and Asia, with the geographical extent of hotspots increasing under a high emissions scenario. These estimates require refinement and a greater knowledge of the abundance, size, feeding regime and location of cattle, and the representation of heat stress should be included in future modelling work. However, our results indicate that the cultivation of more nutritious forage plants and reduced livestock farming in warming regions may reduce this additional source of pastoral greenhouse gas emissions.

Keywords: agriculture, cattle, climate change, fibre, grassland, greenhouse gases, nutrition, protein

## 1. Introduction

Global meat production has increased rapidly in recent years, from 71 million tonnes in 1961 to 318 million tonnes in 2014 (FAOSTAT, 2016). This is due to population growth and a transition to meat-rich diets across many countries (Tilman and Clark, 2014). Grazing lands have expanded to support this production, particularly across Asia and South America, and now cover 35 million km$^2$ of the Earth's surface, with an estimated 1.5 billion cattle, 1.2 billion sheep, 1 billion goats and 0.2 billion buffalo living in livestock production systems (FAOSTAT, 2016). The environmental footprint of supplying meat and

dairy products has increased alongside these rises in human consumption. Livestock farming, including feed production and land use change, enteric sources and manure decomposition produces approximately 7.1 gigatonnes of $CO_2$ and $CO_2$ equivalents annually (GT $CO_2$eq), accounting for 15% of anthropogenic greenhouse gas (GHG) emissions (FAO, 2013). Enteric fermentation by livestock produces 2.8 GT $CO_2$eq of methane each year, with 77 % being produced by cattle (FAO, 2013). The upward trend in livestock production and associated GHG emissions are projected to continue in the future and

global stocks of cattle, goats and sheep are expected to reach 6.3 billion by 2050 (Steinfeld et al., 2006).

Ruminants (cattle and small ruminants such as sheep and goats) consume 80 % (3.7 GT) of the plant material grown to feed livestock (Herrero et al., 2013), and grasses continue to comprise the largest proportion of livestock diets. For example, in the year 2000, 48 % (2.3 billion tons) of the biomass consumed by livestock was grass, followed by grains (1.3 billion tons). The remainder of livestock feed (0.1 billion tons) was the leaves and stalks of field crops, such as corn (maize), sorghum and

soybean (Herrero et al., 2013). The chemical composition and morphology of forage grasses determines their palatability and nutritive value to livestock, thus influencing the amount of feed consumed, efficiency of rumination, rates of weight gain, the quality and volume of milk produced, and reproductive success (Herrero et al., 2015). Forage grasses generally have enhanced nutritive value for livestock if they contain a greater proportion of readily fermentable components such as sugars, organic acids and proteins, and a lower proportion of fibre (Waghorn and Clark, 2004). Furthermore, highly nutritious forage

can reduce ruminant methane production, since feed moves through the digestive system more rapidly (Knapp et al., 2014). Accordingly, regional and inter-annual variability in forage nutritive value generates corresponding variability in the production of meat and dairy products, and variability in the magnitude of ruminant methane emissions (Thornton and Herrero, 2010).

Meat and dairy production in arid, equatorial and tropical regions is often lower than production in temperate regions due to

the lower nutritional quality of forage grasses, a lack of access to inorganic nitrogen (N) fertilisers, infertile soils and adverse climatic conditions (Thornton et al., 2011). Warmer regions are associated with taller, less nutritious and slow-growing grasses with low concentrations of protein, high concentrations of fibre and high plant dry matter content (DM, the proportion of plant dry mass to fresh mass) (Jégo et al., 2013; Waghorn and Clark, 2004). While extremely cold regions are also associated with grasses of low nutritive quality, cold regions are rarely suitable for ruminant livestock (Nielsen et al.,

2013). The timing of grazing and forage harvesting are also important determinants of forage quality. For example, summer harvests frequently produce grasses of lower nutritive quality than spring harvests (Kering et al., 2011). Consequently, grasses of lower forage quality have low dry matter digestibility (DMD, the proportion of plant dry mass which is digestible; high DMD is positively associated with livestock productivity) (Lavorel and Grigulis, 2012; Pontes et al., 2007a). Greater grass nutritive value has been linked to cooler temperatures and N fertiliser addition due to phenological and physiological changes towards delayed flowering, modified stem:leaf ratios, thinner cell walls and reduced lignification, and species turnover (Gardarin et al., 2014; Hirata, 1999; Kering et al., 2011).

Ruminant methane production is calculated using IPCC (2006) methodologies in GHG accounting (Tiers 1,2 and 3), and the more complex methods (Tiers 2 and 3) incorporate the effects of nutritive value (Schils et al., 2007). However, few models have been developed to predict the effects of climate change on forage nutritive value (Kipling et al., 2016), and those which include climate or management have focussed on single livestock species (Jégo et al., 2013) or regions (Graux et al., 2011). Quantifying relationships between forage grass nutritive value, growing conditions and management more broadly, and across many plant species, provides an opportunity to make general projections of future changes to livestock and associated methane production. To our knowledge such relationships have not been systematically assessed at the global scale.

We tested the following hypothesis: that increasing temperatures are associated with grasses of lower nutritive value, delivering higher concentrations of fibre, lower protein and lower DMD with N fertiliser addition having opposite effects. To quantify variation in the nutritive value of forage species growing across a range of bioclimatic zones and to understand the influence of climate and fertiliser application, data were gathered from published literature sources in which field-derived nutritive data were reported. Neutral detergent fibre (NDF, structural plant components; cellulose, lignin and hemicellulose) and crude protein (CP, approximate protein content) are presented as the most commonly reported measurements of forage nutritive value. NDF and CP are generally negatively and positively correlated with livestock productivity, respectively. These data were combined with a range of potentially modifying variables, including temperature, rainfall, rates of N fertiliser addition and photosynthetic pathway. Statistical models were then used to generate projections of future climate induced changes to forage grass nutritive value and cattle methane production.

## 2. Materials and methods

### 2.1. Data acquisition

Data were obtained from peer-reviewed journal articles. Articles were identified by systematically searching the ISI Web of Knowledge (WoK, www.wok.mimas.ac.uk). To avoid researcher bias and to maintain a consistent approach, search terms used to identify articles listed in the WoK were identified *a priori*. Articles were included within the database if nutritive

measurements were related to a specific grass species or hybrid that had been grown in field conditions at a defined location (hereafter termed 'site') and harvested for nutritional analyses at a stated time. Data from experiments conducted in greenhouses or field experiments, i.e. those which manipulated climatic variables, were excluded because the prevailing growing conditions were not representative of the location.

To ensure that the methods for measuring forage nutritive value were consistent across articles, data where included if NDF and CP analyses were carried out on dried samples and presented in units of g/kg DM or % DM. DMD was also recorded when available to test for relationships between NDF, CP and digestibility.

## 2.2. Descriptive data

Descriptive data were included in the database for each data point. These potential explanatory data described the site (latitude, longitude, elevation), experiment (degree of replication, experimental treatments and whether the grassland was a mono- or polyculture), management (fertiliser addition rate, grazing density), soil (type, pH), climate (mean annual temperature [MAT], mean annual rainfall [MAR]), weather during the month of sample collection (mean monthly temperature, total monthly rainfall) as well as data describing the plants photosynthetic pathway system (C3, C4). Data were

recorded from each article from text or tables. When this was not possible, data were obtained from graphs using the digitizing software; Datathief (www.datathief.org).

Sites were allocated to a bioclimatic zone as defined by the Köppen-Geiger Climate Classification system (Kottek et al., 2006) and recorded in the database as arid ($\geq$ 70% of precipitation falls in summer or winter), equatorial (mean temperature of coldest month $\geq$ 18 °C), temperate (mean temperature of warmest month $\geq$ 10 °C and coldest month -3–18 °C) or tundra

(mean temperature of warmest month $\geq$ 10 °C and coldest month $\leq$ -3 °C). The database contained grass nutritive data collected from 32 sites in 16 countries (for detailed site descriptions see Appendix, Table A1) with NDF measurements taken from 35 grass species and CP measurements taken from 46 grass species. Overall, our dataset comprised 803 measurements of nutritive quality taken from 55 different grass species across Asia (11 % of the dataset), Australasia (6 %), Central America (11 %), Europe (34 %), the Middle East (1 %), North America (36 %) and South America (1 %). Our dataset

represented arid (19 % of the dataset, 6 sites), equatorial (1 %, 1 site), temperate (46 %, 16 sites) and tundra (35 %, 9 sites) bioclimatic zones. Across all sites, temperatures at the time of sampling ranged from -5 to 36 °C (MAT: -1–26 °C) and monthly rainfall at the time of sampling ranged from 0.6 to 702 mm (MAR: 38–2378 mm yr$^{-1}$). Data on the rate of N addition were available for 67 % of the dataset, and these rates of fertiliser application ranged from 0 to 357 kg N ha$^{-1}$ yr$^{-1}$. Our dataset represented nutritive values collected from forage plants growing in regions currently suitable for ruminant

livestock.

## 2.3 Gap filling

In many cases data were obtained from the articles analysed, but in some cases there were gaps in the information available. Data most commonly gathered from external sources were weather (sampling temperature and rainfall) and climate (MAT and MAR), which were obtained from the closest weather station to each site, according to the National Centers for Environmental Prediction database (www.ncep.noaa.gov). Weather data for Waimate North was not added to the database because the nearest weather station was 150 km from the site. MAT and MAR were taken as the mean temperature and rainfall over the past ten years. Google Earth (www.earth.google.com) was used to obtain the elevation of the site if this was not stated in the article, based on a digital elevation model.

## 2.4 Statistics

Statistical analyses were carried out using weighted, restricted maximum-likelihood linear mixed-effects (LME) models (Pinheiro and Bates, 2000). Model selection was carried out by including NDF or CP as response variables with multiple potential explanatory variables added as fixed effects to generate full (maximal) models. Fixed effects were mean temperature during the sampling month or MAT, total rainfall during the sampling month or MAR, elevation, rates of N addition and photosynthetic pathway. Grazing density, soil pH and whether the plants were grown in mono- or polyculture were shown not to significantly relate to CP or NDF in LME models in preliminary analyses, and therefore to avoid over-fitting these variables were not included in initial full models (all $P > 0.05$).

For the random effects structure, grass species identity was nested within experimental treatment, and treatments were nested within sites and represented within LMEs, thus accounting for cases where several measurements were taken at the same site, treatment or from the same species. This accounted for differences between species and between sites without making them the focus of our analysis. Any relationships identified therefore included the effects of changes to species identity, and of changes to physiology and phenology. However, a separate model was also fitted for the best represented plant species in the database (*Lolium perenne*) in order to gain an initial insight into the relative roles of physiological response and species turnover. Variation in the sample sizes used to generate treatment means was accounted for by weighting by within-site replication, thus making the influence of a study proportional to its degree of replication (Adams et al., 1997).

Non-significant explantory variables were removed from full models as all terms were found to reduced Akaike's Information Criterion (AIC). The relative influence of each term on model likelihood was assessed by comparing the AIC of the current model with that of a simplified model, with terms deleted until the AIC ceased to decline (Crawley, 2013; Richards, 2005). Temperature and rainfall could not be included together in LME models because these variables were

shown to covary strongly (P < 0.001), so either temperature or rainfall were included in full models based on minimising AIC. LME models were also used to test for relationships between CP and NDF and climate (MAT and MAR), and DMD, and also to test for differences in CP and NDF between bioclimatic zones. For comparison, separate analyses were therefore carried out for MAT or MAR and total rainfall or mean temperature during the month of sampling. All analyses were computed using R, version 3.2.3 (R Development Core Team, 2016).

## 2.5 Enteric methane production modelling

Methane production projections were based on published, experimentally derived relationships between forage NDF content or daily NDF intake (NDFi) and enteric methane production, as measured in cattle. A suite of equations was acquired from published articles with all but one being the product of meta-analysis (Table 1). These equations summarise many measurements of cattle enteric methane production across Africa, Asia, Australasia, Europe, North America and South America, and relate the magnitude of methane production to the nutritive quality of forage and, in some cases, total feed intake. In total, 303 studies were included across these meta-analyses, with methane production measured by hood, mask and whole animal calorimetry, respiration chamber and sulphur hexafluoride ($SF_6$) tracers. Where multiple options were available from a single article, equations were selected for inclusion in our study based on the lowest root mean square prediction error (RMSPE) when this was assessed within the article itself (Moraes et al., 2014; Patra, 2015) or based on the results of a study which compared the accuracy of multiple models in calculating methane production (Appuhamy et al., 2016). These equations, when combined with relationships between forage nutritive quality and temperature identified in this study, were used to model future changes to enteric methane production.

➔ Table 1

NDF and NDFi was calculated using parameters identified by our LME models, which described the relationship between NDF and MAT (see Results), multiplied by estimated daily feed intake or DMI (dry matter intake). Initial modelling based on equations A–E assumed that cattle DMI was 18.8 kg DMI day$^{-1}$, which represents mean DMI across all cattle from North America, Europe and Australasia (Appuhamy et al., 2016). For model F, which represented smaller tropical cattle, a DMI of 7.7 kg day$^{-1}$ was included, which was the mean value presented by Patra (2015). Some equations required values of forage nutritive quality which were not included in this analysis. In these cases, nutritive values were kept constant at 2.8 % dietary fatty acid, 2.8 % ether extract, 162 MJ day$^{-1}$ metabolisable energy intake and 317 MJ day$^{-1}$ gross energy intake, values which were consistent with a range of forage nutritive quality measurements presented elsewhere (e.g. Dalley et al., 1999; Ominski

et al., 2006; Hegarty et al., 2007). These constant values had a lower influence on model outputs than NDF due to their lower absolute values or gradients. To present a range of possible scenarios, estimated changes to methane production were also calculated for a range of DMI values, to represent small, medium and large cattle for the maximum projections (model A), minimum projections (model E) and most variable projections (model F) models. Modelled DMI ranged from 9.7 to 28.9 kg DMI day$^{-1}$ for models A and E (Appuhamy et al., 2016) and from 1.4 to 10.0 kg DMI day$^{-1}$ for model F (Patra, 2015).

Projections of temperature-driven changes to cattle methane production used the HadGEM2 (Hadley Centre Global Environment Model version 2) family of climate models (IPCC, 2014) applying low and high Representative Concentration Pathways (low = RCP 2.6; high = RCP 8.5) to generate geographically explicit estimates of future climate and forage-driven changes to methane production. Projected temperature changes were converted to projected forage-driven changes to enteric methane production for mean sized cattle with mean DMI (as defined above) using a weighted-average model (Table 1), with the relative contribution of the outcomes of equations A–F weighted according to the number of datasets included in each meta-analysis (Adams et al., 1997). The number of measurements used to generate each equation was larger than the number of datasets. For example, Patra (2015) included 142 mean enteric methane values collected from 830 cattle in 35 studies across Australia, Brazil, India and Zimbabwe (Table 1).

HadGEM2 has been identified as a robust model, which is valuable for predictions across climate change scenarios and including biogeochemical feedbacks (Collins et al., 2011). Estimated increases in cattle methane production was calculated as the ratio of methane production based on projected 2050 mean temperatures compared with production based on current temperatures (Hijmans et al., 2005). HadGEM2 models based on RCP 2.6 assumed that GHG mitigation policies are widely adopted, and livestock numbers decline, resulting in a reduction in GHG emissions after 2020. Models based on RCP 8.5 assume that GHG mitigation policies are not adopted, that livestock numbers increase and that GHG emissions continue to increase unabated. RCP 2.6 and RCP 8.5 therefore represented lower and upper projections of future climate and forage-driven increases in cattle methane production. Regions which are unsuitable for ruminant livestock were excluded (Robinson et al., 2014) as were regions which are predicted to exceed 30 °C, since greater temperatures were outside the range of the dataset.

## 3. Results

There was a large range in mean neutral detergent fibre (NDF) across the forage grass species (for a full list of species and a summary of each species nutritive values see Appendix, Table A2), from the lowest, *Pennisetum clandestinum* (46 %) and

*Lolium multiflorum* (46 %) to the highest, *Aristida longiseta* (87 %). There was less variation between the forage grasses in crude protein (CP) (standard deviation of mean CP = 3) than the in NDF (standard deviation of mean NDF = 10). The highest mean CP was recorded in *Pennisetum clandestinum* (23 %) and the lowest recorded from another member of the same genus, *Pennisetum purpureum* (9 %). NDF was correlated strongly with forage dry matter digestibility (DMD), with each 1% increase in NDF linked to a 0.6 % decline in DMD (t = -11.3, P < 0.001). CP was positively related to DMD, however, this significant relationship was dependent upon data from one site. When these outliers were removed there was no significant relationship between CP and DMD (t = -0.2, P > 0.05).

### 3.1 Variation between bioclimatic zones

NDF varied between bioclimatic zones, and grasses growing in cooler temperate or tundra zones had a mean 21 % lower NDF than in warmer arid and equatorial zones (Fig. 1a), but there was no difference between NDF values recorded from arid and equatorial zones. CP also varied between bioclimatic zones, and grasses growing in cooler temperate or tundra zones had a mean of 8 % greater CP than grasses growing in equatorial zones (Fig. 1b). However, there were no differences between the CP contents of grasses growing in arid zones when compared with the other bioclimatic zones.

➔ Figure 1

### 3.2 Environmental determinants of nutritive value

Higher temperatures during the sampling month were associated with increasing NDF across the grasses (Fig. 2) and NDF increased by 0.4 ± 0.06 % (mean ± standard error) for every 1 °C rise in temperature. A small number of samples were collected at very low temperatures (< 0°C) and had low NDF values with a mean of 50 %, whilst at very high temperatures (> 25 °C) NDF values were also high with a mean of 72 %. These extreme values were consistent with the general trends observed. MAT, which represented prevailing climatic conditions rather than sampling conditions, was also positively associated with NDF, but the rate of increase was moderately greater than for sampling temperatures, increasing by 0.9 ± 0.3 % for every 1 °C increase in MAT (Table 2). Rates of N addition were linked to a decline in NDF, with a 100 kg ha$^{-1}$ yr$^{-1}$ increase in the rate of N addition, a moderate rate typical for agricultural grasslands, reducing NDF by 3 ± 1 %. A very high application rate of 350 kg N ha$^{-1}$ yr$^{-1}$ was associated with a decline in NDF of 11 %. These relationships were also tested for *Lolium perenne*, the species best represented in the database. A positive linear relationship was found between NDF and sampling temperature (sites = 20, t = 3.6, P < 0.001), increasing NDF by 13 ± 4 % for every 1 °C increase (over the range 9–

22 °C), and between NDF and MAT (sites = 21, t = 4.6, P < 0.001), increasing NDF by 23 ± 5 % for every 1 °C increase (over the range 6–15 °C). However, there was no relationship between NDF and N for this species.

NDF was also influenced by photosynthetic pathway, with the NDF content of C4 species a mean of 9 % greater than C3 species. These C4 grasses were more commonly recorded at warmer sites, and NDF content was recorded from C4 grasses growing in mean monthly temperatures greater than 15 °C and up to 28 °C whilst NDF was recorded in C3 species growing in temperatures between -5 and 25 °C.

CP was positively related to rates of N addition, with a 100 kg ha$^{-1}$ yr$^{-1}$ increase in the rate of N addition associated with a 2 % increase in CP, and very high application rate of 350 kg N ha-1 yr-1 was associated with a 7 % increase in CP. Mean CP content was 3 % higher for C3 species than for C4 species, but this difference was not significant (P > 0.05). None of the remaining variables were significantly related to CP (all P > 0.05).

→ Figure 2

→ Table 2

## 3.3 Projected future changes to methane production

Applying models A to F to the positive relationship between NDF and MAT resulted in a range of projections for forage and temperature-driven changes to methane production (Fig. 3). Models A to E projected increased methane production with rising temperatures assuming a mean cattle size and DMI, with model A projecting the largest increase in methane production (2.9 % for a 1 °C rise) and model E projecting the lowest increase in methane production for each unit of increased temperature (0.5 % for a 1 °C rise). Models B, C and D produced intermediate values (1.9 %, 1.2 % and 0.7 % for a 1 °C rise, respectively). However, model F projected a reduction in methane production with increased temperatures at mean cattle size (-0.3 % for a 1 °C rise). The models with intermediate predictions (B, C & D), were those based on the largest number of studies (particularly models C and D), and so contributed the most to the weighted mean. Correspondingly, the weighted mean model also projected an intermediate increase in methane production with rising temperatures of 0.9 % for a 1 °C rise in temperatures and 4.5 % for a larger 5 °C rise in temperatures.

→ Figure 3

The effect of simulating changes to cattle size by modifying DMI had contrasting effects across the different models (Fig. 4). In the case of model A, increasing cattle size, consistent with the current global trend towards larger cattle (Herrero et al.,

2013), increased the rise in projected methane production with temperature (0.8–3.7 % for a 1 °C rise, Fig. 4a) whereas larger cattle size decreased the rise in projected methane production for model E (0.3–0.8 % for a 1 °C rise, Fig. 4b). These values represented the largest range of increases in projected methane production with rising temperatures across models A to E. Again, model F behaved differently to the other models; methane production was projected to increase with temperature for the smallest cattle (2.2 % for a 1 °C rise) but decline with temperature for the largest cattle (-1.2 % for a 1 °C

rise, Fig. 4c).

➔  Figure 4

When statistical models were combined with future temperature scenarios, potential hotspots of forage-driven increases in

methane production were identified. The low emissions scenario predicted increases in methane production for mean sized cattle by 1–2 % across most regions, whilst hotpots in North America, Central and Eastern Europe, and Asia saw predicted increases of approximately 3–4 % (Fig. 5a). The high emissions scenario resulted in a larger area experiencing high increases in cattle methane production, with many regions across North and South America, Europe, Central and South Africa, Asia and Australasia increasing by 6–8 % (Fig. 5b). These projections represent estimated change in methane production for each

animal. Simulated decreases and increases in the global cattle inventory are included in climate projections; RCP 2.6 and 8.5, respectively (IPCC, 2014).

➔  Figure 5

**4. Discussion**

Global food consumptions patterns are shifting from traditional diets to diets rich in refined sugars, fats, oils and meats (Tilman and Clark, 2014). Assessments suggest that agricultural GHG emissions need to be reduced by ~1 GT $CO_2$eq annually in order to limit warming to 2 °C above pre-industrial levels by 2100 (Wollenberg et al., 2016). We present preliminary evidence of a previously undescribed positive climate feedback, which may affect our ability to meet these

ambitious GHG emissions targets. Our models project that future temperature-driven reductions in the nutritive value of forage grasses could increase methane production, depending on the emissions scenario, locality and cattle size, thus creating

an additional climate forcing effect. It should be noted, however, that our projections do not incorporate several important but complex factors (for a detailed discussion see Limitations to modelling approach, 4.4), including the effects of climate change on economic growth, technological uptake and land availability, which have not been fully quantified (Audsley et al., 2014; Havlík et al., 2014). Nevertheless, the potential magnitude of future decreases in grass nutritive value and corresponding increases in methane production means that these projections cannot be ignored, and are identified here as a research area requiring careful future work and refinement.

### 4.1 Variation in nutritive and functional traits

Forage grass nutritive value varied substantially, between- and within-species, and across bioclimatic zones, with our data indicating that 34–90 % of the dry weight of the grass that livestock consume is fibre and 5–36 % is protein. These ranges are greater than those presented elsewhere, for example NDF has been shown to range from 35–67 % (O'Donovan et al., 2011) and CP from 14–24 % across several European grass species and cultivars (Pontes et al., 2007b), but these greater ranges are to be expected given the wider biogeographic coverage of our study.

NDF values were generally higher and CP generally lower in warmer bioclimatic zones than in cooler zones, and this is likely to be one reason why livestock productivity is lower across arid, equatorial and tropical regions. Reduced nutritive value in these zones may be driven by increased abundances of plants with adaptations to prevent heat stress and avoid water loss; such as greater stem:leaf ratios, narrowly spaced veins, greater hair densities, thicker cell walls, a higher proportion of epidermis, bundle sheath, sclerenchyma and vascular tissues, and greater concentrations of lignin and silica (Kering et al., 2011). The C4 photosynthetic pathway is also an adaptation to heat and water stress and C4 plants were more commonly recorded in warmer conditions than C3 plants, and C4 plants were also associated with lower nutritive value. This is in line with studies that have measured elevated enteric methane production in cattle consuming high fibre C4 grasses compared with those consuming C3 grasses (Ulyatt et al., 2002). Across warmer bioclimatic zones reduced forage nutritive values may be driven by increased abundances of C4 species, and of taller, slow growing species with a conservative growth strategy (Martin and Isaac, 2015; Wood et al., 2015). Large variation within- and between-species highlights the potential for the cultivation and breeding of grasses to enhance livestock nutrition, which may promote resistance to future environmental changes.

### 4.2 Relationships between nutritive value, environment and management

NDF was positively related to temperatures at the time of sampling and MAT. Links between higher temperatures and declining nutritive values, and between declining nutritive values and increasing enteric methane production have been

established under controlled condition (Knapp et al., 2014). Our results indicate that the same mechanisms may operate at a global scale. MAT represents prevailing climatic conditions, and elevated NDF is likely driven by a shift towards grasses with heat and drought stress adaptations, and conservative functional traits associated with slow growth (Gardarin et al., 2014). The positive relationship between sampling temperature and NDF may also be linked with changes to phenology, such as advanced flowering dates and rapid tissue aging (Hirata, 1999). The timing of measurements may also have played a role in increasing NDF, since later harvests generally produce grasses of lower nutritive quality (Kering et al., 2011). Temperature driven reductions in forage grass nutritive value is consistent with mechanistic and empirical models (Barrett et al., 2005; Kipling et al., 2016). However, our results contrast with a meta-analysis of temperature manipulation experiments, which did not reveal any relationships between warming and nutritive value, although this study was across a relatively small temperature gradient (Dumont et al., 2015). The relationships between forage nutritive value and both sampling temperatures and MAT imply that species compositional (i.e. turnover in species identity), phenological and physiological changes each play a role. Patterns generated by these different processes were not directly disentangled in our study. However, there were relationships between both MAT and sampling temperatures, and NDF, when measured from one species, *Lolium perenne*. This pattern will likely have been driven by changes to physiology, phenology and harvesting time, but not species turnover. The effect size when only *L.perenne* was included in our analysis was larger than for all plant species, though it was over a smaller temperature range of 6 to 15 °C. This large response indicates that phenological and physiological changes may play a significant role in driving the reduction of NDF under warming, and that changes may occur without species turnover. The positive relationships between NDF, and both MAT and sampling temperatures, across species and within an individual species, provide additional evidence that our projections are robust.

N fertiliser addition generally increases the productivity of grasslands, since the greater majority of these ecosystems are N limited (LeBauer and Treseder, 2008; Lee et al., 2010). We present data which suggests that N addition may also increase grass nutritive value, decreasing NDF by around 3–11 % (low to high fertiliser application rates), and with to an associated increase in CP by 2–7 %. Increased rates of N addition has been linked previously to increased abundances of grass species with 'fast' functional traits, with reduced fibre and increased protein content (Pontes et al., 2007a). N addition did not alter nutritive quality for *L.perenne* and therefore the relationship between N and NDF for all species could represent species turnover, rather than changes to physiology or phenology. N addition could partially offset the negative effects of warmer temperatures on forage grass nutritive value in polyculture (where there is species turnover), although N enrichment may also have other, potentially unwanted, ecosystem impacts (Manning, 2012). Improved nutrition management by farmers may also partially offset predicted gains in enteric methane production (Caro et al., 2016).

### 4.3 Explorations of future methane production

Our estimates suggest that future cattle enteric methane production may change by a mean weighted value of 0.9 % (-0.3–2.9 %) for an initial 1 °C increase in temperatures, assuming no change in mean cattle size, due to reduced forage nutritive quality. This increase would translate to an annual change in methane production across the global cattle inventory of approximately 0.02 GT $CO_2$eq (-0.01–0.06 GT $CO_2$eq). With a larger 5 °C increase in temperatures the projected change in cattle methane production of 4.5 % (-2–14 %) translates to a global change of approximately 0.09 GT $CO_2$eq (-0.02–0.3 GT $CO_2$eq). However, it may be the case that some areas may fall outside the range of our models under greater warming (i.e. those with MAT > 30 °C) increasing the uncertainty of these estimates. Since RCPs already include simulated changes in the global cattle inventory (IPCC, 2014), our results demonstrate that forage and temperature-driven increases in methane production may offset some of the methane reductions assumed to come from fewer cattle (RCP 2.6) or may further amplify methane increases from a greater number of cattle (RCP 8.5). Whether methane production will change towards the mean, upper or lower end of the projected ranges is clearly dependent on which model is correct. We postulate that the most likely models are model C, which represented North American cattle, and the mean weighted model, as these included the largest number of studies (thus representing a large range of cattle sizes and breeds). Both gave comparable and intermediate outputs. Five of the six models were consistent with studies linking increased forage fibre with greater enteric methane production (e.g. Moraes et al., 2014), and therefore estimated increased methane production under warmer temperatures. One model (model F) projected declines in future enteric methane production with temperature. However, care must be taken in this case as the model was parameterised using data collected from smaller animals and across tropical regions. When the smallest animals were simulated with this model, as is consistent with smaller tropical breeds such as Zebu, enteric methane was also projected to rise with temperatures.

The trend towards larger cattle across many regions could also influence the magnitude of changes to enteric methane production, because larger cattle have greater feed and fibre intakes (Knapp et al., 2014). Model predictions for larger animals were more variable and therefore both the magnitude of emissions and the uncertainty surrounding these estimates increases with cattle size. The magnitude of projected change across the different models was also dependent on whether NDF or DMI was the dominant term. Furthermore, our projections are limited to cattle. However, there is emerging evidence that reductions in the nutritive value of forage also leads to increased enteric methane production from sheep (Ramin and Huhtanen, 2013) and buffalo (Patra, 2014). Together cattle, buffalo and sheep contribute >95 % of global GHG emissions from enteric fermentation (FAO, 2013) and if our projections hold across the global ruminant inventory then overall enteric methane production will increase to a greater magnitude than we predict. Our calculations are also limited to cattle that consume grass. We therefore do not account for the trend towards permanently housed cattle, particularly across Europe and North America. This may further increase emissions because the mixed diets of housed cattle increase enteric methane production by around 58 % (March et al., 2014; O'Neill et al., 2011).

Hotspots of future increases in enteric methane production were identified across North America, Central and Eastern Europe, and Asia using a low GHG emissions scenario combined with our weighted mean model. Hotspots became more widespread, and of greater magnitude, in a high GHG emissions scenario. At present the greatest densities of cattle can be found in parts of Asia, North and South America, Europe and across Australasia (FAOSTAT, 2016), and many of these regions are projected to experience the greatest forage nutrition-driven increase in cattle methane production. Added to this, meat production has increased by 3.6 % across Africa and 3.4 % across Asia over the past decade, compared with a 1 % increase across Europe (FAOSTAT, 2016), indicating greater future growth across these regions. Losses in forage quality could drive farmers into more extensive farming systems across many regions, because larger land areas will be required for each animal. Therefore, it may be necessary to limit the growth of livestock production systems in warmer and drier regions, particularly those likely to experience future warming, if significant losses in livestock production efficiency and increases in methane emissions are to be avoided.

Cattle methane production can be reduced by growing more nutritious forage plants, N fertiliser addition, feed supplements (e.g. macroalgae and fats), adjusting rumen pH, increased concentrate feeding, genetic selection, and feeding methane inhibitors (Duin et al., 2016; Machado et al., 2014). However, implementing many of these measures is not feasible at a global scale, is unlikely to result in sufficient reductions in GHG emissions to meet ambitious GHG reduction targets, and may also promote other negative environmental effects such as biodiversity loss, nitrous oxide emissions and pollution to air and water (Manning, 2012; Wollenberg et al., 2016). Ruminant meats (beef and lamb) produce around 250 times greater GHG emissions per gram of protein than legumes (crops from the family Leguminosae); and eggs, seafood, aquaculture, poultry and pork all have lower emissions than ruminant meats (Tilman and Clark, 2014). A global switch in human diets and a transition to more sustainable agricultural practices, as well as a greater prevalence of organic and silvopastoral farming, may reduce our reliance on intensively farmed cattle and other ruminants. In countries with high or increasing meat consumption, these measures could reduce the environmental impacts of agriculture, contribute to GHG emissions cuts, and with an associated improvement in human health (Springmann et al., 2016).

## 4.4 Limitations to modelling approach

There are many uncertainties associated with modelling plant and livestock systems and all of the relavent factors could not be considered in our analysis. Future attemps to refine our predictions therefore require additional processes to be represented mechanistically and data to parameterise these processes (Hill et al., 2016). Current livestock models require many inputs, which are not universally available, and do not account for variation across all individuals, breeds and regions. Furthermore, current mechanistic vegetation models do not quantitatively consider climate-driven changes to forage nutritive quality (Kipling et al., 2016). Recent work has addressed knowledge gaps in empirical models, such as quantifying methane

produced by cattle across Africa and other tropical regions, thus improving the coverage of these models (Jaurena et al.,

2015; Patra, 2015). However, there continues to be low geographic coverage of forage quality data in equatorial and tropical

regions, where the nutritive quality of forage is typically lower than temperate regions (Nielsen et al., 2013). Furthermore,

the effects of heat stress on enteric methane production has not been fully quantified (Kadzere et al., 2002) and the

anticipated near-doubling of the global livestock inventory was also not included in our projections, because future changes

in the distribution of cattle and technological advances are currently unknown (Herrero et al., 2015). If livestock numbers

increase in rapidly warming regions then we predict that there will be an associated rise in enteric methane production.

Increased grazing pressure may also alter plant species composition, thus changing the nutritive value and extent of grazing

lands (Gardarin et al., 2014). Other global environmental changes, such as elevated $CO_2$ (Barbehenn et al., 2004; Roumet et

al., 1999), and increased frequency of drought, flooding and extreme weather events could also affect methane production

(Hoover et al., 2014).

## 5. Conclusions

We present preliminary evidence of future temperature-driven declines in forage nutritive quality and corresponding

increases in enteric methane production. Upscaling the GHG footprint of the current livestock inventory to the 2050

projected inventory increases annual GHG emissions from enteric sources from 2.8 GT $CO_2$eq to 4.7 GT $CO_2$eq. However,

our projections reveal that the geographical distribution of livestock, changes to their size and diet and the interactions

between nutritive values, climate and fertilisers may modify the GHG footprint of cattle. The incorporation of a greater

number of factors which were not included in our analysis, along with more detailed measures of how forage quality changes

across environmental gradients would help to refine our estimates. Nevertheless, our projections reveal robust general trends

and highlight a potentially important and previously unrecognised climate change feedback, with important implications for

GHG emissions targets, future warming, agricultural policies and food security.

## 6. Author contribution

445     M. Lee and P. Manning designed the approach and M. Lee carried out data collection and analyses. M. Lee developed

predictive models and maps. M. Lee prepared the manuscript with contributions from all co-authors.

## 7. Acknowledgements

This paper was produced following consultation with the members of RBG Kew Plant Nutrition and Traits Database steering committee. Thanks to Gerhard Boenisch, Jens Kattge, Charlie Marsh and Alex Papadopulos for editorial advice and discussions on presentation and analyses. The authors would like to thank two anonymous reviewers for comments which improved the manuscript.

## 8. Competing interests

The authors declare that they have no conflict of interest.

## 9. Data availability

Data can be obtained by contacting the lead author directly. Some of our data has been obtained from journals which are not open access and cannot be freely distributed.

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

Table 1: A summary of the published equations used to model grass nutritive quality driven changes in methane production,
giving details of cattle type (D = dairy, B = beef), regions covered (AF = Africa, AS = Asia, AUNZ = Austalia and New Zealand, EU = Europe, NA = North America, SA = South America) and the number of studies included in each analysis. Root mean square prediction error (RMSPE) values are also presented.

| Ref* | Cattle | Regions | Studies | Equation (CH$_4$ =)** | RMSPE*** | Model |
|------|--------|---------|---------|-----------------------|----------|-------|
| 1 | D | AS | 1 | $5.1 \times NDF^2 - 39.3 \times NDF + 360.0$ | - | A |
| 2 | D | EU, NA, AUNZ | 21 | $-2.8 + 3.7 \times NDFi$ | 18.3 | B |
| 3 | D,B | NA | 172 | $1.6 + 0.04 \times MEi + 1.5 \times NDFi$ | 17.9 | C |
| 4 | D,B | NA | 62 | $0.2 + 0.04 \times GEi + 0.1 \times NDF - 0.3 \times EE$ | 17.9 | D |
| 5 | D | EU | 12 | $1.2 \times DMI - 1.5 \times FA + 0.1 \times NDF$ | 16.9 | E |
| 6 | D,B | AF, AS, AUNZ, SA | 35 | $-1.0 + 0.3 \times DMI + 0.04 \times DMI^2 + 2.4 \times NDFi - 0.3 \times NDFi^2$ | 31.4 | F |

* 1. Kasuya and Takahashi, 2010, 2. Storlien et al., 2014, 3. Ellis et al., 2007, 4. Moraes et al., 2014, 5. Nielsen et al., 2013, 6. Patra, 2015

** NDF = neutral detergent fibre (%DM), NDFi = neutral detergent fibre intake (kg day$^{-1}$), MEi = metabolisable energy intake (MJ day$^{-1}$), GEi = gross energy intake (MJ day$^{-1}$), EE = dietary ether extract (%DM), DMI = dry matter intake (kg day$^{-1}$) and FA = dietary fatty acid (%DM)

*** As presented by Appuhamy et al (2016) except ref 4 and 6 which were presented within the referenced article

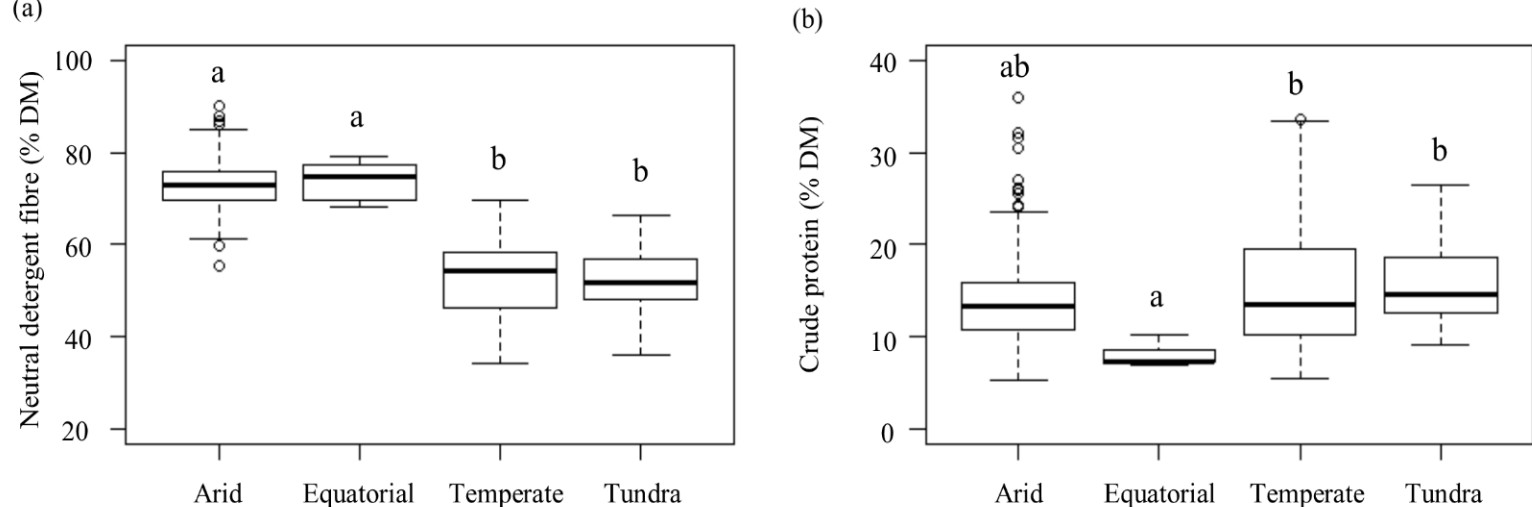

Figure 1: Boxplots of (a) the neutral detergent fibre (NDF) and (b) the crude protein (CP) content of grasses located in bioclimatic zones as described by the Köppen-Geiger Climate Classification system. Significant differences between zones, as identified by LME models, are denoted by different letters (P < 0.05).

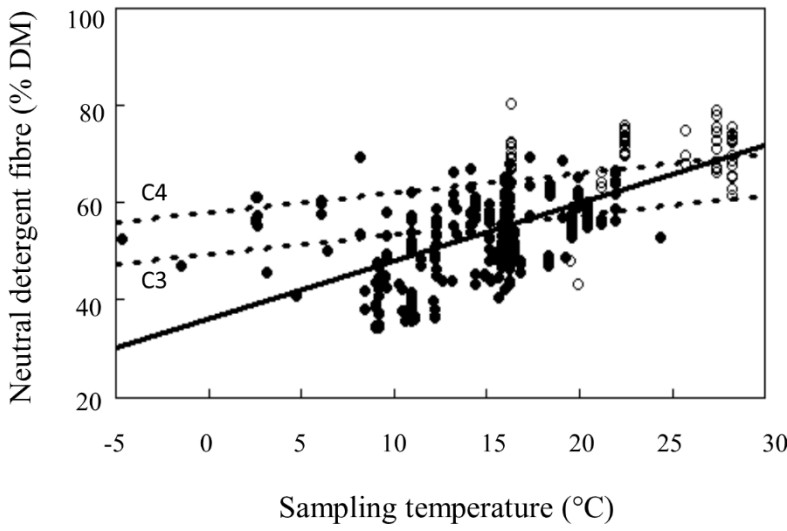


Figure 2: Linear relationship between forage neutral detergent fibre (NDF) content and temperature (°C) at the time of sampling. Filled circles are C3 species and open circles are C4 species ($P < 0.05$). Dotted lines represent best fit lines for C3 NDF = 0.4T + 49) and C4 species (NDF = 0.4T + 58). The continuous line represents the best fit line for all species excluding other factors included within LME (NDF = 1.1T + 36).


Table 2: Minimum adequate linear mixed effects models for forage neutral detergent fibre (NDF) and crude protein (CP). Values represent slopes except C4 pathway values which represent absolute differences between C3 pathway (Intercept) and C4 pathway. Site numbers differ between response types since temperature at the time of sampling and both NDF and CP were not always available from all articles.


| Response | Sites | Factor | Value | SE | DF | T | P |
|---|---|---|---|---|---|---|---|
| NDF | 20 | Intercept | 49.4 | 2.0 | 287 | 25.3 | <0.001 |
| | | Temperature at sampling (°C) | 0.4 | 0.06 | 287 | 5.8 | <0.001 |
| | | N addition (kg N ha$^{-1}$ yr$^{-1}$) | -0.03 | 0.01 | 287 | -3.4 | <0.001 |
| | | C4 pathway presence | 8.7 | 3.2 | 33 | 2.7 | <0.05 |
| NDF | 32 | Intercept | 43.4 | 3.7 | 300 | 11.6 | <0.001 |
| | | MAT (°C) | 0.9 | 0.3 | 19 | 3.8 | <0.01 |
| CP | 25 | Intercept | 14.2 | 1.0 | 484 | 14.8 | <0.001 |
| | | Rainfall (mm mth$^{-1}$) | -0.002 | 0.002 | 484 | -0.8 | 0.43 |
| | | N addition (kg N ha$^{-1}$ yr$^{-1}$) | 0.02 | 0.006 | 484 | 3.0 | <0.01 |
| | | C4 pathway presence | -2.9 | 1.7 | 46 | -1.7 | 0.1 |
| CP | 27 | Intercept | 15.9 | 1.6 | 575 | 9.9 | <0.001 |
| | | MAR (mm yr$^{-1}$) | -0.001 | 0.001 | 24 | -0.5 | 0.65 |

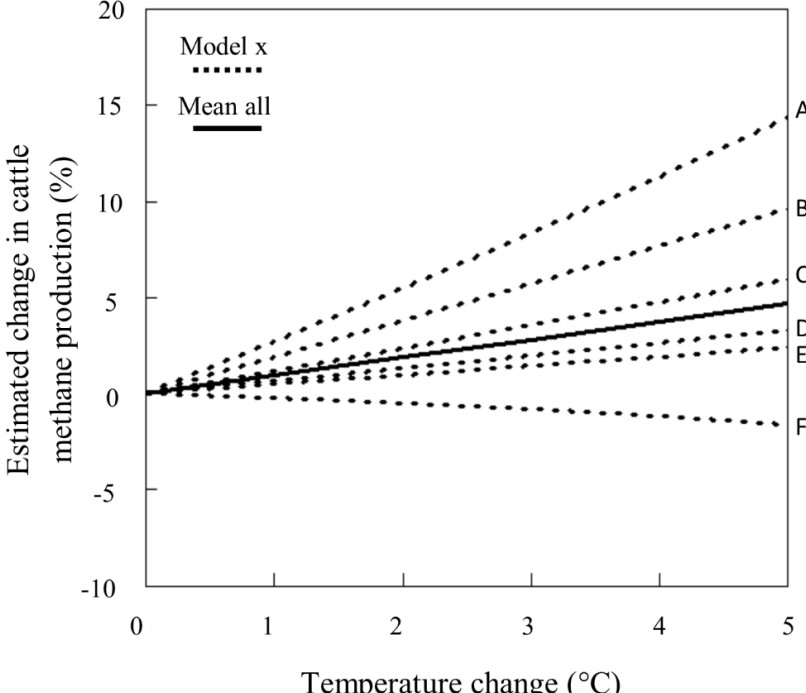

Figure 3: Estimated change in cattle methane production with temperature derived declines in grass nutritive quality. Dotted lines represent six model outputs as defined by equations A – F (defining relationships between grass nutritive quality and methane production) when combined with the inverse relationship between temperature and grass nutritive quality presented in this article. The continuous line represents the mean weighted model; mean methane production predicted by all six equations, weighted by the number of contributing datasets.


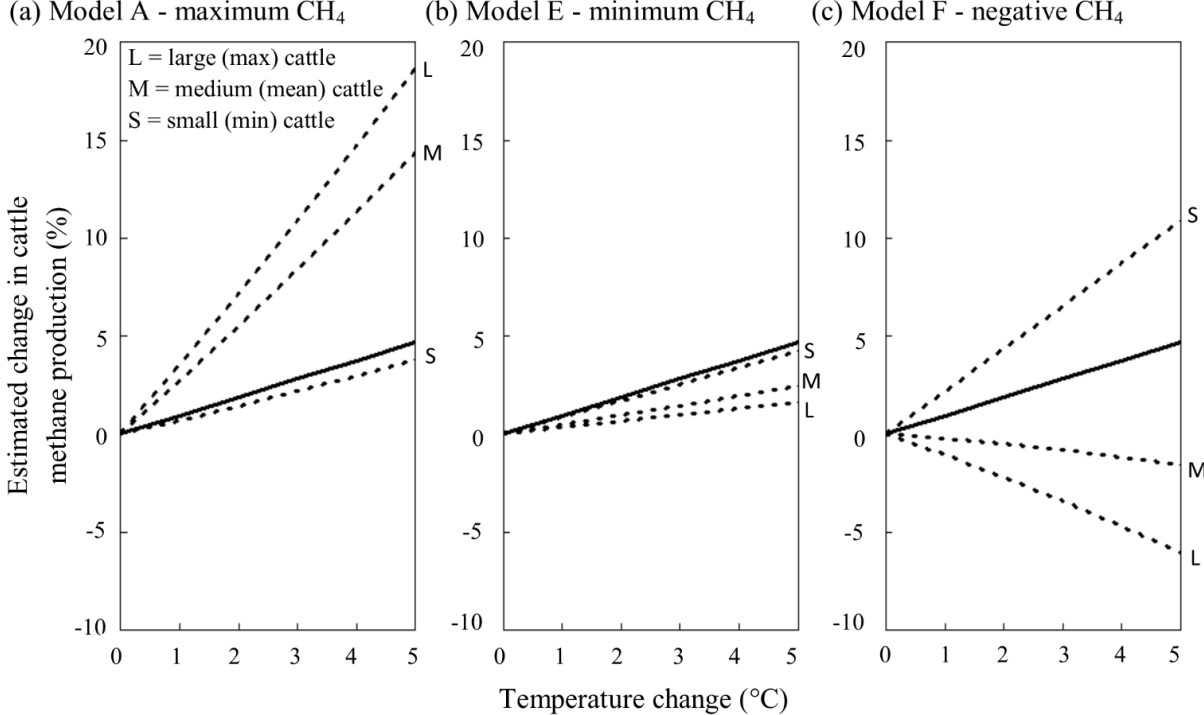


Figure 4: Estimated change in enteric methane production with temperature change for (a) the model predicting the largest increases in methane production (maximum $CH_4$), (b) the model predicting the lowest increase in methane production (minimum $CH_4$) and (c) the model which predicts both increases and decreases in methane production (negative $CH_4$). Dotted lines represent predictions for minimum sized (small, S), mean sized (medium, M) and maximum sized (large, L)

cattle. S, M and L cattle were defined as cattle consuming 9.7, 18.8 and 28.9 kg DMI day$^{-1}$, respectively, for model A and E. S, M and L cattle were defined as consuming 1.4, 7.7 and 10 kg DMI day$^{-1}$, respectively, for model F which represents less productive tropical regions. Values do not include projected changes to the global cattle inventory.

(a)

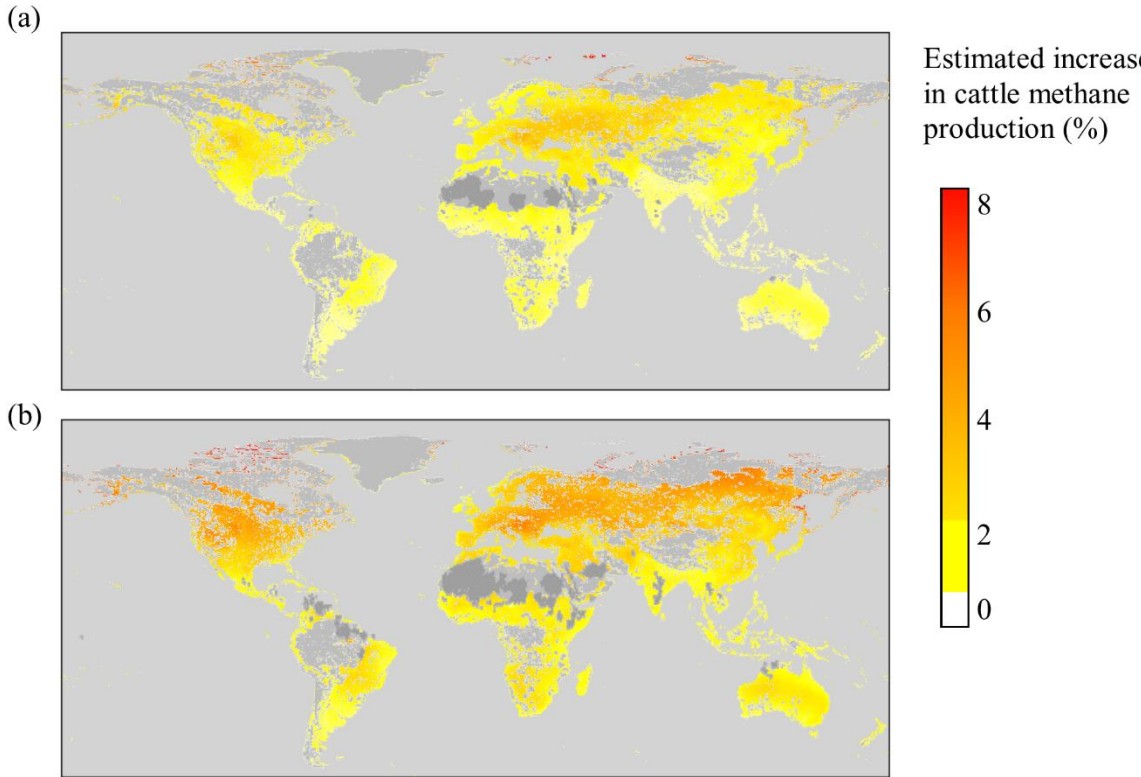

Estimated increase
in cattle methane
production (%)

Figure 5: Predictions of climate and forage-driven increases in cattle methane production (%) under 2050 predicted temperatures using (a) a low estimate of future temperature changes (RCP 2.6) and (b) a high estimate of future temperature changes (RCP 8.5). Regions in dark grey are currently unsuitable for ruminant livestock and regions with predicted temperatures greater than 30 °C are shaded black since they are beyond the range of the dataset. Values do not include projected increases in the global cattle inventory which are simulated to decrease and increase in RCP 2.6 and 8.5, respectively. Estimates are based on HADGEM2 climate projections combined with the mean weighted model. Estimates are based on data collected from arid (19 %), equatorial (1 %), temperate (46 %) and tundra (35 %) bioclimatic zones, all currently suitable for ruminant livestock.

**Appendix**

Table A1: Sites included in the database, detailing latitude, longitude, Mean Annual Temperature (MAT, °C), Mean Annual Rainfall (MAR, mm) and altitude (m). The site with no climatic data is indicted by an em-rule (–). Some sites did not contribute both NDF and CP values.


| Site | Country | Latitude | Longitude | MAT | MAR | Altitude |
|------|---------|----------|-----------|-----|-----|----------|
| Calden[1] | Argentina | -38.450 | -63.750 | 15.0 | 400.0 | 95 |
| Buenos Aires[2] | Argentina | -37.183 | -62.133 | 15.9 | 602.7 | 181 |
| Mutdapily[3] | Australia | -27.767 | 152.667 | 19.9 | 815.0 | 40 |
| Pernambuco[4] | Brazil | -8.014 | -34.951 | 25.7 | 2310.3 | 23 |
| Lacombe[5] | Canada | 52.467 | -113.733 | 2.4 | 466.0 | 855 |
| Melfort[5] | Canada | 52.817 | -104.600 | 0.7 | 439.0 | 483 |
| Alberta[6] | Canada | 53.756 | -113.339 | 3.0 | 455.8 | 674 |
| Fredericton[7] | Canada | 45.917 | -66.604 | 5.6 | 1065.0 | 26 |
| Gansu[8] | China | 37.667 | 103.533 | -1.0 | 385.7 | 3000 |
| Fodder Research[9] | Czech Republic | 49.517 | 15.967 | 6.9 | 617.0 | 560 |
| Grange[10,11,12,13] | Ireland | 53.500 | -6.670 | 6.3 | 877.3 | 83 |
| Moorepark[14] | Ireland | 52.163 | -8.260 | 10.0 | 1040.0 | 70 |
| Tohoku[15] | Japan | 39.733 | 141.133 | 9.3 | 1180.0 | 110 |
| Ohda[16] | Japan | 35.167 | 132.500 | 15.9 | 1603.9 | 53 |
| Sumiyoshi[17] | Japan | 31.983 | 131.467 | 17.3 | 2378.0 | 11 |
| Nuevo Leon[18] | Mexico | 25.717 | -100.033 | 22.0 | 500.0 | 393 |
| Sauces Ranch[19] | Mexico | 25.407 | -99.776 | 22.0 | 360.0 | 272 |
| Chifeng[20] | Mongolia | 42.261 | 118.931 | 4.5 | 380.0 | 900 |
| Wageningen[21] | Netherlands | 51.967 | 5.667 | 9.3 | 771.4 | 7 |
| Lincoln[22] | New Zealand | -43.633 | 172.467 | 11.5 | 581.2 | 22 |
| Waimate North[23] | New Zealand | -35.300 | 173.900 | – | – | 83 |
| Quassim[24] | Saudi Arabia | 26.308 | 43.767 | 24.7 | 160.6 | 652 |
| Alpine region[25] | Slovenia | 46.050 | 14.467 | 10.8 | 914.8 | 300 |
| Atatürk[26] | Turkey | 39.917 | 41.267 | 4.4 | 37.9 | 1850 |
| Black Sea[27] | Turkey | 41.244 | 36.510 | 14.6 | 709.3 | 4 |
| Erzurum[28] | Turkey | 39.906 | 41.271 | 5.7 | 409.4 | 1905 |
| Aberystwyth[29] | United Kingdom | 52.367 | -4.083 | 10.0 | 1174.0 | 100 |
| Ty Gwyn[30] | United Kingdom | 52.267 | -4.083 | 10.0 | 1823.8 | 257 |
| Fort Keogh[31] | United States | 46.367 | -105.083 | 8.2 | 498.3 | 719 |
| Ithaca[32] | United States | 42.440 | -76.500 | 8.4 | 963.9 | 120 |
| Logan[33] | United States | 41.767 | -111.817 | 9.1 | 509.6 | 1406 |
| Mount Pleasant[32] | United States | 41.110 | -73.810 | 11.5 | 1327.0 | 100 |

[1] Distel et al., 2005, [2] Catanese et al., 2009, [3]Callow et al., 2003, [4]dos Santos et al., 2003, [5]McCartney et al., 2008, [6]Suleiman et al., 1999, [7]Bélanger and Mcqueen, 1997, [8]Dong et al., 2003, [9]Skladanka et al., 2010, [10]Conaghan et al., 2008[11]Keating and O'Kiely, 2000, [12]King et al., 2012, [13]Mceniry et al., 2014, [14]Beecher et al., 2015, [15] Nashiki et al., 2005, [16]Kobayashi et al., 2008, [17]Hirata et al., 2008, [18,19]Ramirez, 2007, [20]Zhao et al., 2012, [21]Smit et al., 2005, [22]Bryant et al.,


2012, [23]Ulyatt et al., 2002, [24]Al-Ghumaiz and Motawei, 2011, [25]Čop et al., 2009, [26]Akgun et al., 2008, [27] Surmen et al., 2013, [28]Sahin et al., 2012, [29]Lee et al., 2001, [30]Weller and Cooper, 2001, [31]Haferkamp and Grings, 2002, [32]Cherney and Cherney, 1997, [33]Griggs et al., 2007


Table A2: Species included in the database showing NDF (% DM) and CP (%DM) mean, standard deviation (SD), maximum (Max) and minimum (Min) values. Hybridised species are denoted by a multiplication sign (x). The site with no climatic data is indicted by an em-rule (–).

| | NDF (% DM) | | | | CP (% DM) | | | |
|---|---|---|---|---|---|---|---|---|
| | Mean | SD | Max | Min | Mean | SD | Max | Min |
| *Agropyron cristatum* | – | – | – | – | 17 | 7 | 36 | 8 |
| *Agropyron intermedium* | – | – | – | – | 16 | 5 | 26 | 9 |
| *Agropyron riparium* | – | – | – | – | 16 | 3 | 23 | 11 |
| *Agropyron trachycaulum* | – | – | – | – | 15 | 5 | 25 | 10 |
| *Agropyron trichophorum* | – | – | – | – | 16 | 5 | 27 | 11 |
| *Alopecurus pratensis* | 58 | 9 | 70 | 39 | 15 | 4 | 24 | 8 |
| *Aristida longiseta* | 87 | 1 | 88 | 85 | – | – | – | – |
| *Arrhenatherum elatius* | 61 | 1 | 61 | 60 | 8 | 1 | 9 | 7 |
| *Bouteloua curtipendula* | 74 | 3 | 79 | 72 | 11 | 3 | 14 | 8 |
| *Bouteloua gracilis* | 83 | 5 | 90 | 77 | – | – | – | – |
| *Bouteloua trifida* | 74 | 3 | 76 | 70 | 11 | 4 | 15 | 8 |
| *Brachiaria brizantha* | 75 | – | 75 | 75 | 7 | – | 7 | 7 |
| *Brachiaria fasciculata* | 64 | 5 | 72 | 60 | 14 | 4 | 18 | 10 |
| *Bromus inermis* | – | – | – | – | 16 | 6 | 26 | 7 |
| *Cenchrus ciliaris* | 76 | 2 | 78 | 74 | – | – | – | – |
| *Cenchrus incertus* | 77 | 3 | 80 | 74 | – | – | – | – |
| *Chloris ciliata* | 70 | 3 | 72 | 65 | 13 | 3 | 18 | 10 |
| *Dactylis glomerata* | 58 | 5 | 64 | 43 | 14 | 4 | 26 | 9 |
| *Digitaria insularis* | 72 | 2 | 75 | 70 | 11 | 3 | 13 | 7 |
| *Echinochloa crusgalli* | 64 | 2 | 66 | 63 | 11 | 1 | 12 | 10 |
| *Elymus nutans* | – | – | – | – | 14 | 1 | 15 | 13 |
| *Elymus sibiricus* | – | – | – | – | 14 | 8 | 26 | 5 |
| *Elytrigia intermediata* | – | – | – | – | 20 | 9 | 32 | 6 |
| *Eremochloa ophiuroides* | – | – | – | – | 12 | 3 | 20 | 8 |
| *Festuca arundinacea* | 57 | 3 | 60 | 53 | 15 | 4 | 23 | 9 |
| *Festuca arundinacea* x *Lolium multiflorum* | 58 | 2 | 61 | 56 | 8 | 1 | 9 | 8 |
| *Festuca pratensis* | – | – | – | – | 11 | 1 | 12 | 11 |
| *Festuca rubra* | – | – | – | – | 17 | 3 | 21 | 11 |
| *Hilaria belangeri* | 79 | 4 | 83 | 75 | – | – | – | – |
| *Holcus lanatus* | 54 | 9 | 65 | 39 | 11 | 4 | 19 | 5 |
| *Hordeum brevisubulatum* | – | – | – | – | 14 | 1 | 15 | 13 |
| *Leptochloa filiformis* | 70 | 4 | 75 | 67 | 12 | 2 | 15 | 10 |
| *Lolium multiflorum* | 46 | 6 | 56 | 36 | 15 | 5 | 28 | 6 |

| | | | | | | | | |
|---|---|---|---|---|---|---|---|---|
| *Lolium multiflorum × Festuca pratensis* | – | – | – | – | 12 | 1 | 13 | 12 |
| *Lolium perenne* | 50 | 8 | 62 | 34 | 18 | 8 | 34 | 7 |
| *Lolium perenne × Festuca pratensis* | – | – | – | – | 11 | 0 | 11 | 10 |
| *Panicum hallii* | 71 | 3 | 76 | 67 | 13 | 5 | 18 | 8 |
| *Panicum obtusum* | 65 | 8 | 74 | 55 | 14 | 2 | 17 | 12 |
| *Pascopyrum smithii* | – | – | – | – | 18 | 6 | 26 | 7 |
| *Paspalum notatum* | – | – | – | – | 12 | 3 | 19 | 9 |
| *Paspalum unispicatum* | 68 | 2 | 70 | 64 | 11 | 3 | 13 | 9 |
| *Pennisetum clandestinum* | 46 | 4 | 48 | 43 | 23 | 1 | 23 | 22 |
| *Pennisetum maximum* | 78 | 1 | 79 | 77 | 7 | 0 | 7 | 7 |
| *Pennisetum purpureum* | 69 | 1 | 70 | 68 | 9 | 1 | 10 | 9 |
| *Phalaris arundinacea* | 58 | 5 | 67 | 52 | – | – | – | – |
| *Phleum pratense* | 51 | 8 | 67 | 36 | 15 | 4 | 23 | 9 |
| *Poa crymophila* | – | – | – | – | 13 | 5 | 20 | 8 |
| *Rhynchelytrum repens* | 72 | 2 | 74 | 69 | 10 | 2 | 11 | 7 |
| *Roegneria turczaninovii* | – | – | – | – | 15 | 1 | 16 | 14 |
| *Setaria grisebachii* | 72 | 8 | 81 | 61 | 14 | 4 | 17 | 9 |
| *Setaria macrostachya* | 74 | 7 | 86 | 63 | 13 | 2 | 16 | 11 |
| *Stipa clarazii* | 55 | 2 | 57 | 54 | 16 | 6 | 22 | 11 |
| *Stipa eriostachya* | 66 | 6 | 69 | 59 | 10 | 5 | 16 | 7 |
| *Tridens eragrostoides* | 73 | 2 | 76 | 71 | 13 | 2 | 17 | 11 |
| *Tridens muticus* | 75 | 3 | 78 | 72 | 11 | 4 | 16 | 8 |
