# Peer review of "Forage quality declines with rising temperatures, with implications for livestock production and methane emissions"

_Biogeosciences, 2016_

## Referee Comment (RC1) · Anonymous Referee #1 · 25 Oct 2016

General Comments

The authors present a meta-analysis of forage studies in order to ascertain any impacts of growing conditions on methane production by livestock. Overall I think this study is a valuable contribution by highlighting a positive feedback between temperature increases and methane production by livestock. The clear and succinct project raises many important questions for global methane contributions and feedbacks under future climate scenarios, but could be improved by some additional considerations and clarifications. I did not find that the analysis of Nitrogen addition to add much value as part of the results section, and because of the limited data on this part of the study (and lack of a significant impact for the species with the most data) is to preliminary

an analysis for inclusion. In addition, clarification and discussion if these results are indicative of changes within a species or between species (and relative contributions of each) is needed as this is considerable factor in the assumptions for their model. Furthermore, projections related to RCP 2.6 and 8.5 and increased methane should be developed further than currently presented to clarify the relationship to livestock assumptions in these models and present the spatial variability between regions of the world in more detail. Finally, additional clarification of other assumptions and limitations to this study are needed to generate discussion and thoughts about taking these coarse projections further. Grassland communities are complicated and although the authors show a response to general long term temperature to forage nutritive value, interannual and geographic variability (plus management) are additional important factors. More detailed comments on specifics sections of the manuscript follow.

Specific Comments

Line 42, if 48% of the biomass is grass, would be good to know what composes the other 52%. This is a big deal for methane production and would help with conclusions and discussion points.

Line 51, consider talking about tundra regions here as well since you base your results on this climate type. Since these are harsh climate do they behave like arid regions (stressful) or temperate regions (cooler, so greater nutritive value)?

Line 84, you need to talk about the size of the database here and not just percentages. It is important to know the distribution and number of species across climate types, the amount of data that your fertilizer model is based off of, etc. It is hard to determine if the results you have are from within species variability or across species variability. The two lead to different conclusions and are an important to discussing changes in methane production from cattle in the same locations (are we assuming a change in forage species?).

Line 91, a brief discussion of whether harvested time impacts DM and other variables

and then later on, account for this in the analyses (i.e. on line 109 it is reported that a sample was taken at -5 degrees C).

Line 143, is this for all temperature and rainfall values? Both the month of collection and mean annual values?

Line 167, RCP 2.6 and RCP 8.5 incorporate projections in the amount of livestock as a part of determining changes to radiative forcing. Be explicit here that you are restraining your analysis to just the projected temperature changes as determined by RCP 2.6 and then 8.5, not any changes related to projections in number of livestock in the scenarios or any assumptions about where, feed type, etc.

Line 201, consider splitting the relationship between C3 and C4 plants here as you do in the model later on. Looks like a different response but hard to tell.

Line 203, please revise the table caption to better reflect the four models presented. The comparison to the results section and why the numbers of sites differ, plus the two models for NDF and CP are not clear.

Line 223, please clarify the figure explanation, it is hard to determine where the two scenarios come from in your temperature model for each size of livestock. Also consider some clarification in the methods section where you present equations for these (line 150).

Line 223, I find the nitrogen addition discussion distracting and not needed for the main part of this paper. I think you could make a great point focusing on temperature and save discussion of nitrogen addition to the discussion. It complicates the methods section (data collection) and this is a small part of your database (8%), plus you find a temperature impact for the main species in your data, but not a nitrogen effect (making this a more complicated question).

Line 235, I like the analyses but the figures presented could be more informative. In this case these figures mainly represent areas with larger projected temperature change.

Consider some alternative presentation, such as presenting the % change by continent, or other factor. A table or figure that presents changes by geographic location for different sizes of cattle would give much more information than currently presented in the text and figure. You could even consider ramifications of increased numbers of livestock in addition to the temperature impacts (as referred to in the discussion but not presented in the results).

Line 253, talk here a bit more about the assumptions in the model you have created (data sources, species variability vs community variability, forage type, etc.). Again, I think this is a valuable study and addition, just need to explain what additional information is needed to go beyond the "coarse projections."

Line 300, what is the magnitude difference of increased methane in housed cattle vs. the increase of methane from grass at warmer temperatures. Can you say your overall projection may increase?

Line 331, I liked the discussion overall, and think you cover a lot of good points about the conclusions of the study. Two additional factors to consider are the unknowns of the impact of increased $CO_2$ on NDF and CP for grass species (especially C3), how would this impact your conclusions. And secondly, consider a discussion about grazing pressure (which I know you excluded) changing community composition and species response, and those impacts to CD and NDF.

Technical corrections

Line 117, units wrong on elevation (not likely km).

---

## Referee Comment (RC2) · Anonymous Referee #2 · 30 Oct 2016

General comments The study aims at investigating the relationship between forage quality, methane emissions from livestock, and projected future emissions. The topic is interesting, relevant and timely. The authors have done a good job gathering data to show the variability of forage quality for key quality parameters, plant species and across world climates. And that in itself would be useful material to be published (e.g. Fig. 1 and 2, Table 1) in a specialized forage science journal. What I find less robust, is the use of statistical models derived with forage quality data, and the temperature under which the forage was sampled, to make (future) predictions of methane emissions by livestock. The analyses that would make this manuscript relevant for Biogeosciences are based on a few equations (derived from statistical analyses) which related methane

emissions to the quality of the feed. Temperature is an explanatory variable that was used by other authors (Hirotaka Kasuya and Junichi Takahashi – see Asian-Aust. J. Anim. Sci. Vol. 23, No. 5: 563 - 566) to explain the intake of NDF, whereas methane emissions are driven by the intake of NDF. I find the extrapolation of these equations too week to make global predictions of methane emissions. This sort of study, interesting and relevant, would be better substantiated using vegetation models that represent the physiological processes through which temperature would affect feed quality, and live-stock models that would describe the effect of temperature on livestock (heat stress?) affecting the emission of methane. There are more weaknesses in the assumptions used for the study, which I describe below under specific comments. Unfortunately, I don't find this manuscript suitable for publication in Biogeosciences.

Specific comments L108: the authors used temperature at time of sampling, mean annual temperature (MAT) and monthly rainfall (MAR) over the past 10 years. The quality of the forage is associate to the current growing season, most like a seasonal and cumulative effect. So the use of an average long term (10 years) temperature of the temperature of the month of sampling seem inappropriate as predictors of feed quality.

L143-148: the use of equations developed for one experiment conducted in Japan, with a limited set of feedstuff (only 4 temperate climate species) to extrapolate global methane emissions seem largely inadequate for the purpose.

L192: I would have expected a species effect in the analyses of NDF. Under the same climate and soil there will be plants with largely different values of NDF, and other quality parameters simply because of genetic differences.

L232: the use of the selected statistical models derived from one single experiment, with future temperatures seem inappropriate to predict both future and actual methane emissions globally.

L247: I disagree with the authors. They don't describe here a climate feedback, but an

artifact of the use of statistical models and projected temperatures. The relationship between temperatures and plant quality parameters is largely known in ecology. That explains the differences between ecotypes across the globe. However, the authors extend these relationships to the calculation of methane emissions, and that seems incorrect. L264: the differences in NDF and CP across climate doesn't mean that ruminants are under nutritional stress. Livestock keepers manage different species and breed adapted to their climate across the globe. And therefore it is not correct to use one equation derived for Bos taurus dry cows in Japan to predict global emissions of ruminants.

---

## Author Comment (AC1) · 25 Nov 2016

**Author response to both reviewers': Forage quality declines with rising temperatures, with implications for livestock production and methane emissions**

**Mark A. Lee, Aaron P. Davis, Mizeck G. G. Chagunda, Peter Manning**

We would like to thank both reviewers for taking the time to read our manuscript and provide us with thought provoking and well-structured reviews. We agree with Reviewer 1, that we have raised some important questions regarding global methane contributions and feedbacks, and we agree with Reviewer 2, that the topic is interesting, relevant and timely. We are also pleased that the reviewers felt that we had done a good job gathering data to show the variability of forage quality for key quality parameters and plant species, across world climates. We collected a large dataset from 33 published articles from 32 sites in 17 countries (Table A1). This resulted in over 800 rows of data, each with detailed climatic, edaphic and management variables from 55 fodder plant species, and this dataset allowed us to identify general relationships (Table A2).

Reviewer 2 highlights the limitations of our modelling approach, in particular the parameters which we did not include. While we agree with the reviewer that some potentially important aspects are not represented in our model, it was not our intention to deliver a definitive estimate of the magnitude of the climate-forage-livestock emissions feedback, as this would require significant model development and is beyond the scope of this paper. Instead we hoped to highlight this logically robust and potentially important process, and support it with empirical evidence. We will clarify this point in the revised manuscript. We also hope that this paper will inspire discussion and a more complete work programme, in which all of the potentially important parameters can be modelled in more detail. We agree that we should highlight the parameters we did not model and then discuss their implications in more detail in the Discussion section.

One area where the reviewers' disagree is in the predictions of future methane production resulting from our analyses. Whilst Reviewer 1 would like to see our predictions presented in greater geographical detail, Reviewer 2 felt that the relationship between forage quality and elevated methane production is not sufficient to make robust predictions. We agree with the concerns of Reviewer 2. To rectify this, we would repeat our predictions using several other published models. We would use three models: one representing North American cattle (Ellis et al., 2007; Moraes et al., 2014); a second representing European cattle (Nielsen et al., 2013); and a third model representing North American, European and Australasian cattle (Storlein et al, 2014). As summarised in a recent review paper, all models which include neutral detergent fibre (NDF) and which are derived from *in-vivo* studies of forage dietary composition show that NDF has a positive effect on methane production in cattle (Appuhamy et al., 2016). Based on our observations of the model parameters we believe that these further models will provide broad support for our predictions and it is highly unlikely that the relationship is an artefact as Reviewer 2 postulates. However, it is likely that this portfolio of model outputs will differ in the magnitude of predicted methane production and so we propose to include a comparison of model outputs in our final paper.

We believe that the all of the remaining reviewers' comments can be satisfactorily addressed or clarified in the final manuscript, and that doing so will result in a significantly improved manuscript that allays their concerns. We also feel that our conclusions are robust and that this paper is of high interest to the readership of Biogeosciences.

Responses to specific comments are presented below, with reviewer comments in blue and author responses in black.

**Responses to Reviewer 1**

General comments: The authors present a meta-analysis of forage studies in order to ascertain any impacts of growing conditions on methane production by livestock. Overall I think this study is a valuable contribution by highlighting a positive feedback between temperature increases and methane production by livestock. The clear and succinct project raises many important questions for global methane contributions and feedbacks under future climate scenarios, but could be improved by some additional considerations and clarifications.

Response: We thank reviewer 1 for these positive comments and, in particular, that we have raised some important questions regarding global methane contributions and feedbacks.

Proposed changes: No changes requested by reviewer.

I did not find that the analysis of Nitrogen addition to add much value as part of the results section, and because of the limited data on this part of the study (and lack of a significant impact for the species with the most data) is too preliminary an analysis for inclusion.

Response: We feel that it is vital to include nitrogen in our analyses since it is a major global change driver and widely known to have an important effect on forage productivity (LeBauer and Treseder, 2008; Lee et al., 2010) and plant tissue chemistry (Aerts and Decaluwe, 1994; Aerts, 2009; Cherney and Cherney, 1997; Kering et al., 2011; King et al., 2012). Of the 803 rows of data we collected for this study, 535 rows included data on the quantity of nitrogen added, around 67% of the dataset.

Proposed changes: Although we feel that the nitrogen material does not distract from the main message of the paper and adds an important element to it, we would be happy to remove this section if the Editor sees fit. We will also clarify in the text the large amount of fertiliser application rate data that was collected if it remains part of the manuscript.

In addition, clarification and discussion if these results are indicative of changes within a species or between species (and relative contributions of each) is needed as this is considerable factor in the assumptions for their model.

Response: There will be substantial variation between species responses to climate and nitrogen, as supported by the data that we have presented. However, we were not focussed on these differences, preferring to focus on the effects of climate and fertilizer management. We therefore included species and site as random effects in our mixed effects statistical modelling. By doing this we accounted for differences between species and between sites without making them the focus of our analysis. Patterns generated by compositional and physiological changes could not be disentangled in our study, only the response of both aspects to mean annual temperatures (MAT). However, the positive responses to monthly temperatures and MAT imply that both compositional and physiological changes each play a role.

Proposed changes: We will add more detail in the manuscript so that these methodological aspects are clearer.

Furthermore, projections related to RCP 2.6 and 8.5 and increased methane should be developed further than currently presented to clarify the relationship to livestock assumptions in these models and present the spatial variability between regions of the world in more detail.

Response: We are pleased that the reviewer felt that our models are informative and we would like to present their outputs in greater detail. However, given the strongly conflicting opinions of Reviewers 1 and 2 we would like guidance from the Editor. We believe that our models are robust.

However, to demonstrate the potential uncertainty in our predictions and to reduce our reliance on a single published study, which quantified the relationship between forage NDF and methane production (Kasuya and Takahashi, 2010), we would now like to recalculate our predictions using a suite of other published models.

Proposed changes: We have summarised this proposed approach in more detail above. Based on our observations of the model parameters we believe that these additional models will broadly support our published model. There will likely be some quantitative differences between model outputs and we will include a comparison between models in our final paper. If these models provide consistent predictions we could then justifiably include more detailed maps.

Finally, additional clarification of other assumptions and limitations to this study are needed to generate discussion and thoughts about taking these coarse projections further. Grassland communities are complicated and although the authors show a response to general long term temperature to forage nutritive value, inter-annual and geographic variability (plus management) are additional important factors.

Response: We agree with this comment.

Proposed changes: We would address this by including more detail about the assumptions and limitations of our work. This would also enhance our manuscript in generating discussion and further research. Further detail of these changes is given below.

More detailed comments on specifics sections of the manuscript follow. Specific Comments Line 42, if 48% of the biomass is grass, would be good to know what composes the other 52%. This is a big deal for methane production and would help with conclusions and discussion points.

Response: Pasture contributes 48% (2.3 billion tons) of the biomass consumed by livestock, followed by grains (1.3 billion tons, 28%). The remainder is from leaves and stalks of field crops, such as corn (maize), sorghum or soybean (Herrero et al., 2013). This would certainly help with our discussion and we would like to include this.

Proposed changes: Manuscript text will be updated to include this information.

Line 51, consider talking about tundra regions here as well since you base your results on this climate type. Since these are harsh climate do they behave like arid regions (stressful) or temperate regions (cooler, so greater nutritive value)?

Response: We would be happy to discuss tundra regions and feel this would improve the paper.

Proposed changes: A short section on tundra would be added.

Line 84, you need to talk about the size of the database here and not just percentages. It is important to know the distribution and number of species across climate types, the amount of data that your fertilizer model is based off of, etc. It is hard to determine if the results you have are from within species variability or across species variability. The two lead to different conclusions and are an important to discussing changes in methane production from cattle in the same locations (are we assuming a change in forage species?).

Response: We agree that this is essential for the reader and apologise for the lack of information presented in the manuscript. We summarise the size of the dataset above.

Proposed changes: We will provide further detail of how this data breaks down across biogeographic regions, species, functional types etc. in the revised manuscript.

Line 91, a brief discussion of whether harvested time impacts DM and other variables and then later on, account for this in the analyses (i.e. on line 109 it is reported that a sample was taken at -5 degrees C).

Response: We would be happy to include this. This will also improve our introduction and discussion.

Proposed changes: Manuscript to be updated to include this information.

Line 143, is this for all temperature and rainfall values? Both the month of collection and mean annual values?

Response: Separate analyses were carried out for monthly and annual values.

Proposed changes: We will add detail to clarify this and also include a comparison of the results.

Line 167, RCP 2.6 and RCP 8.5 incorporate projections in the amount of livestock as a part of determining changes to radiative forcing. Be explicit here that you are restraining your analysis to just the projected temperature changes as determined by RCP 2.6 and then 8.5, not any changes related to projections in number of livestock in the scenarios or any assumptions about where, feed type, etc.

Response: This is correct and we are happy to clarify this.

Proposed changes: Manuscript will be updated to make this point clear.

Line 201, consider splitting the relationship between C3 and C4 plants here as you do in the model later on. Looks like a different response but hard to tell.

Response: We will split this in the figure. We do demonstrate in the results that C3 and C4 plants differ, which is an interesting finding from our study.

Line 203, please revise the table caption to better reflect the four models presented. The comparison to the results section and why the numbers of sites differ, plus the two models for NDF and CP are not clear.

Response: Thank you for this suggestion.

Proposed changes: The revised paper will include this additional information for the current models and those to be added.

Line 223, please clarify the figure explanation, it is hard to determine where the two scenarios come from in your temperature model for each size of livestock. Also consider some clarification in the methods section where you present equations for these (line 150).

Response: We agree that these changes will help clarify the paper

Proposed changes: We will clarify these points and update the methods section accordingly.

Line 223, I find the nitrogen addition discussion distracting and not needed for the main part of this paper. I think you could make a great point focusing on temperature and save discussion of nitrogen addition to the discussion. It complicates the methods section (data collection) and this is a small part of your database (8%), plus you find a temperature impact for the main species in your data, but not a nitrogen effect (making this a more complicated question).

Response: We feel that it is vital to include nitrogen in our analyses. Of the 803 rows of data we collected for this study, 535 rows included data on the quantity of nitrogen added. This is much

greater (67%) than the 8% suggested above. We have discussed this comment in greater detail above.

Proposed changes: We await the recommendations of the Editor but suggest that the nitrogen component is retained.

Line 235, I like the analyses but the figures presented could be more informative. In this case these figures mainly represent areas with larger projected temperature change. Consider some alternative presentation, such as presenting the % change by continent, or other factor. A table or figure that presents changes by geographic location for different sizes of cattle would give much more information than currently presented in the text and figure. You could even consider ramifications of increased numbers of livestock in addition to the temperature impacts (as referred to in the discussion but not presented in the results).

Response: If the Editor approves this approach we would produce maps and tables which show predictions for specific regions. We limited our discussion to the ramifications of increases in the overall number of livestock, as the location of future increases in livestock production is highly uncertain and we therefore excluded this from our discussion.

Proposed changes: We would like to re-run our predictions using models based on studies of North American, European and Australasian cattle and we can therefore provide additional detail on geographical variation. We will discuss the issue of increased numbers of livestock. We will also clarify in the text that geographically explicit predictions of future changes in livestock number, species and breeds would be required to refine these predictions further.

Line 253, talk here a bit more about the assumptions in the model you have created (data sources, species variability vs community variability, forage type, etc.). Again, I think this is a valuable study and addition, just need to explain what additional information is needed to go beyond the "coarse projections."

Response: We agree with this comment and thank the reviewer for their positive comments.

Proposed changes: We would include more information on our assumptions in the methods and also provide a discussion of these issues.

Line 300, what is the magnitude difference of increased methane in housed cattle vs. the increase of methane from grass at warmer temperatures. Can you say your overall projection may increase?

Response: The methane emissions of housed cattle, which are fed a mixed ration of silage, grass and other components are generally much greater than those reared outdoors (O'Neill et al., 2011).

Proposed changes: We will present the magnitude of this difference in the revised manuscript.

Line 331, I liked the discussion overall, and think you cover a lot of good points about the conclusions of the study. Two additional factors to consider are the unknowns of the impact of increased C02 on NDF and CP for grass species (especially C3), how would this impact your conclusions. And secondly, consider a discussion about grazing pressure (which I know you excluded) changing community composition and species response, and those impacts to CD and NDF.

Response: Thank you for your positive and useful comments.

Proposed changes: We are aware of several studies of the effects of $CO_2$ on plant nutritive quality and we would include these within our Discussion (e.g. Barbehenn et al., 2004; Roumet et al., 1999).

**Responses to Reviewer 2**

General comments: The study aims at investigating the relationship between forage quality, methane emissions from livestock, and projected future emissions. The topic is interesting, relevant and timely. The authors have done a good job gathering data to show the variability of forage quality for key quality parameters, plant species and across world climates. And that in itself would be useful material to be published (e.g. Fig. 1 and 2, Table 1) in a specialized forage science journal.

Response: We thank Reviewer 2 for these positive comments. We agree that we have collated a large dataset as summarised above.

Proposed changes: None required.

What I find less robust, is the use of statistical models derived with forage quality data, and the temperature under which the forage was sampled, to make (future) predictions of methane emissions by livestock. The analyses that would make this manuscript relevant for Biogeosciences are based on a few equations (derived from statistical analyses) which related methane emissions to the quality of the feed. Temperature is an explanatory variable that was used by other authors (Hirotaka Kasuya and Junichi Takahashi – see Asian-Aust. J. Anim. Sci. Vol. 23, No. 5: 563 - 566) to explain the intake of NDF, whereas methane emissions are driven by the intake of NDF. I find the extrapolation of these equations too week to make global predictions of methane emissions.

Response: The reviewers' comments do highlight some of the uncertainties in our analysis. However, while we agree with the reviewer that some potentially important aspects of the relationship are not represented in out model, it was not our intention to deliver a definitive estimate of the climate-forage-livestock emissions feedback. This would require significant model development and is beyond the scope of this paper. Instead we hoped to highlight this logically robust and potentially important process, and support it with empirical evidence. We agree that making predictions based on this one study is a major limitation of our study, but one which can be rectified.

Proposed changes: We have presented a detailed summary of our proposed solution above, i.e. to re-run our predictions using several other published models.

This sort of study, interesting and relevant, would be better substantiated using vegetation models that represent the physiological processes through which temperature would affect feed quality, and livestock models that would describe the effect of temperature on livestock (heat stress?) affecting the emission of methane. There are more weaknesses in the assumptions used for the study, which I describe below under specific comments. Unfortunately, I don't find this manuscript suitable for publication in Biogeosciences.

Response: We agree that vegetation models would offer a useful alternative methodology by way of comparison. However, we are not aware of any vegetation model which is suitable for this kind of analysis and on this scale, as the key vegetation properties that drive livestock methane emissions (e.g. NDF) are not represented in current models. We are also not aware of any complimentary livestock models that would be suitable. Incorporating these elements into existing models is a major task, albeit one which we hope to stimulate with this paper. Also, although we greatly appreciate the value of simulation models, we also know that they have their own limitations and uncertainties, making empirical studies a vital complement and source of information. Again, we would like to highlight that our aim with this paper is to present a potentially important process, which we believe is supported by empirical evidence. We hope that this manuscript inspires future work of the type the reviewer suggests.

Proposed changes: None suggested.

Specific comments L108: the authors used temperature at time of sampling, mean annual temperature (MAT) and monthly rainfall (MAR) over the past 10 years. The quality of the forage is associate to the current growing season, most like a seasonal and cumulative effect. So the use of an average long term (10 years) temperature of the temperature of the month of sampling seem inappropriate as predictors of feed quality.

Response: The use of MAT in this context allows us to link our statistical models to future climate models, which present predictions in terms of MAT. We agree that temperature during the month of sampling, which we have presented, and temperature during the growing season, which we have not presented, could also be valuable predictors. For the former, our output was very similar in terms of both gradient and intercept and this gave us additional confidence that our predictions are robust.

Proposed changes: We would be happy to compare the effects of MAT and growing temperatures in a new analysis, and then compare these values with those that we have used. We will present these either as a result or as a sensitivity analysis in an appendix.

L143-148: the use of equations developed for one experiment conducted in Japan, with a limited set of feedstuff (only 4 temperate climate species) to extrapolate global methane emissions seem largely inadequate for the purpose.

Response: This comment has been addressed previously. We will now calculate our predictions using a suite of models.

Proposed changes: As above.

L192: I would have expected a species effect in the analyses of NDF. Under the same climate and soil there will be plants with largely different values of NDF, and other quality parameters simply because of genetic differences.

Response: This point was also raised by reviewer 1 and our response to it can be found above. There will be substantial variation in species responses to climate and nitrogen, as supported by the data that we have presented. Please see our response above.

Proposed changes: We apologise that this point was not clear enough and will add detail to the manuscript, particularly in quantifying the random effect generated by species identity.

L232: the use of the selected statistical models derived from one single experiment, with future temperatures seem inappropriate to predict both future and actual methane emissions globally.

Response: As stated above, we intent to re-run our predictions using a suite of other models. However, we would like to point out that our statistical models were based on a dataset generated from many published articles.

Proposed changes: We will make the preliminary/discovery nature of our study more explicit throughout the manuscript, with caveats and cautions where necessary.

L247: I disagree with the authors. They don't describe here a climate feedback, but an artifact of the use of statistical models and projected temperatures. The relationship between temperatures and plant quality parameters is largely known in ecology. That explains the differences between ecotypes across the globe. However, the authors extend these relationships to the calculation of methane emissions, and that seems incorrect.

Response: We do not believe this is a statistical artefact. The logic is robust and the relationships presented are supported by previous published work. If the relationship between temperatures and plant quality is largely known and the relationship between plant quality and methane production well resolved then we believe that the overarching concept we have identified is robust. As summarised in a recent review paper, elevated forage NDF has been demonstrated to have a positive effect on methane production in cattle (Appuhamy et al., 2016).

Proposed changes: None suggested.

L264: the differences in NDF and CP across climate doesn't mean that ruminants are under nutritional stress. Livestock keepers manage different species and breed adapted to their climate across the globe.. And therefore it is not correct to use one equation derived for Bos taurus dry cows in Japan to predict global emissions of ruminants

Response: Whether different breeds of ruminants are under nutritional stress is outside the scope of our study. We identified that increasing temperatures are likely to reduce the nutritive value of forage grasses. It would be interesting for a further study to examine the effects of climate on nutritional stress in different species and breeds. However, we still believe that it is reasonable to propose that forage of reduced nutritive value may increase the prevalence of nutritional stress in livestock.

Proposed changes: We will discuss nutritional stress as part of a section highlighting the limitations of our study. We will also highlight that nutritional stress will vary between livestock breeds and species. Again, we have discussed our proposed solution to the issue of relying on one equation above.

**References continue on the page below.**

**References**

Aerts, R.: Nitrogen supply effects on leaf dynamics and nutrient input into the soil of plant species in a sub-arctic tundra ecosystem, Polar Biol., 32(2), 207–214, doi:10.1007/s00300-008-0521-1, 2009.

Aerts, R. and Decaluwe, H.: Effects of Nitrogen Supply on Canopy Structure and Leaf Nitrogen Distribution in Carex Species, Ecology, 75(5), 1482–1490 [online] Available from: ISI:A1994NV66200026, 1994.

Appuhamy, J. A. D. R. N., France, J. and Kebreab, E.: Models for predicting enteric methane emissions from dairy cows in North America, Europe, and Australia and New Zealand, Glob. Chang. Biol., TBC, doi:10.1111/gcb.13339, 2016.

Barbehenn, R. V., Chen, Z., Karowe, D. N. and Spickard, A.: C3 grasses have higher nutritional quality than C4 grasses under ambient and elevated atmospheric CO2, Glob. Chang. Biol., 10(9), 1565–1575, doi:10.1111/j.1365-2486.2004.00833.x, 2004.

Cherney, D. J. R. and Cherney, J. H.: Grass forage quality and digestion kinetics as influenced by nitrogen fertilization and maturity, J. Appl. Anim. Res., 11(2), 105–120, doi:10.1080/09712119.1997.9706170, 1997.

Ellis, J. L., Kebreab, E., Odongo, N. E., McBride, B. W., Okine, E. K. and France, J.: Prediction of methane production from dairy and beef cattle, J. Dairy Sci., 90(7), 3456–3466, doi:10.3168/jds.2006-675, 2007.

Herrero, M., Havlík, P., Valin, H., Notenbaert, A., Rufino, M. C., Thornton, P. K., Blümmel, M., Weiss, F., Grace, D. and Obersteiner, M.: Biomass use, production, feed efficiencies, and greenhouse gas emissions from global livestock systems., Proc. Natl. Acad. Sci. U. S. A., 110(52), 20888–93, doi:10.1073/pnas.1308149110, 2013.

Kasuya, H. and Takahashi, J.: Methane emissions from dry cows fed grass or legume silage, Asian-Australasian J. Anim. Sci., 23(5), 563–566, doi:10.5713/ajas.2010.90488, 2010.

Kering, M. K., Guretzky, J., Funderburg, E. and Mosali, J.: Effect of Nitrogen Fertilizer Rate and Harvest Season on Forage Yield, Quality, and Macronutrient Concentrations in Midland Bermuda Grass, Commun. Soil Sci. Plant Anal., 42, 1958–1971, doi:10.1080/00103624.2011.591470, 2011.

King, C., McEniry, J., Richardson, M. and O'Kiely, P.: Yield and chemical composition of five common grassland species in response to nitrogen fertiliser application and phenological growth stage, Acta Agric. Scand. Sect. B-Soil Plant Sci., 62(March 2016), 644–658, doi:10.1080/09064710.2012.687055, 2012.

LeBauer, D. and Treseder, K.: Nitrogen limitation of net primary productivity in terrestrial ecosystems is globally distributed, Ecology, 89(2), 371–379, 2008.

Lee, M., Manning, P., Rist, J., Power, S. A. and Marsh, C.: A global comparison of grassland biomass responses to CO2 and nitrogen enrichment., Philos. Trans. R. Soc. Lond. B. Biol. Sci., 365(1549), 2047–2056, doi:10.1098/rstb.2010.0028, 2010.

Moraes, L. E., Strathe, A. B., Fadel, J. G., Casper, D. P. and Kebreab, E.: Prediction of enteric methane emissions from cattle, Glob. Chang. Biol., 20(7), 2140–2148, doi:10.1111/gcb.12471, 2014.

Nielsen, N. I., Volden, H., Åkerlind, M., Brask, M., Hellwing, a. L. F., Storlien, T. and Bertilsson, J.: A prediction equation for enteric methane emission from dairy cows for use in NorFor, Acta Agric. Scand. Sect. A - Anim. Sci., 63(3), 126–130, doi:10.1080/09064702.2013.851275, 2013.

O'Neill, B. F., Deighton, M. H., O'Loughlin, B. M., Mulligan, F. J., Boland, T. M., O'Donovan, M. and Lewis, E.: Effects of a perennial ryegrass diet or total mixed ration diet offered to spring-calving

Holstein-Friesian dairy cows on methane emissions, dry matter intake, and milk production., J. Dairy Sci., 94, 1941–1951, doi:10.3168/jds.2010-3361, 2011.

Roumet, C., Laurent, G. and Roy, J.: Leaf structure and chemical composition as affected by elevated CO2: Genotypic responses of two perennial grasses, New Phytol., 143(1), 73–81, doi:10.1046/j.1469-8137.1999.00437.x, 1999.

Storlein, T.M., Volden, H., Almoy, T., Beauchemin, K.A., Mcallister, T.A., Harstad, O.M.: Prediction of enteric methane production from dairy cows. Acta Agric. Scand. Sect. A - Anim. Sci, 64, 98–109. 2014.

---

## Author Response (AR1)

Royal Botanic Gardens, Kew Richmond

Surrey

TW9 3AB

27 January 2017

Dear Professor Stoy,

We would like to re-submit a revised version of our article entitled "Forage quality declines with rising temperatures, with implications for livestock production and methane emissions" to your journal for review. We would like to thank both Reviewers and yourself for taking the time to read our article and provide us with detailed, thought-provoking and constructive comments. We feel that this has greatly enhanced the manuscript. We hope that it is now considered suitable for publication in Biogeosciences.

Please see our responses to your comments and both Reviewers' comments presented below. Original comments are in blue and our responses are in black.

Yours faithfully,

marklee

Dr Mark Lee (on behalf of all authors)

**Editor comments**

The recommendation instead is to use a plant physiological model to quantify the mechanisms by which temperature impacts forage quality. The challenge is that livestock are critical in the climate system and must be studied post haste, yet only basic studies of ruminant physiology oftentimes exist. (I'm running into similar challenges with bison, where it appears that only one calorimeter study on a one-year-old female cow has been published to date.)

My recommendation is to reconsider this manuscript after major revisions. It is timely and well-written, an important gap in the literature has been noted, and projections were qualified as 'coarse', which I assume to mean 'approximate'. At the same time, shifting the focus of the manuscript to important uncertainties in plant and ruminant physiology while noting that the proposed feedback exists when extrapolating the results of current understanding would result in a more conservative analysis that creates a clear path for future studies.

'Coarse' is qualitative, and I'm not sure what it means either. Using an 'if/then' framework in which global projections hold if current understanding is robust would be a more nuanced way to communicate this potentially quite important feedback to the scientific community.

Please submit a revised version of the manuscript that addresses referee comments and adjusts the tone of the manuscript to focus more on gaps in the literature and a bit less on global extrapolations and I would be happy to reconsider it for publication in Biogeosciences.

**Author response**

Thank you for your positive comments. We agree that this is a timely paper and we hope that it will inspire discussion as well as future work that improves the accuracy of our projections. We have now made a number of changes to the manuscript and also included several new analyses which we feel has improved the manuscript greatly. Most notably, we no longer rely on one published equation to project enteric methane production from the relationship between grass nutritive quality and rising temperatures. Instead we have rerun our analyses using six published models and present the range of possible outcomes from these models as well as a mean model, weighted by the number of contributing studies. This will give the reader additional confidence that the results are robust and that the important feedback which we have identified is likely.

We have also reduced the focus on global extrapolations and revised our projections downwards, to a more conservative set of projections based on our mean weighted model. The revised manuscript has a very small proportion of the text dedicated to geographically explicit projections or global values for increases in enteric methane production. However, we do still include one map and a small number of global projections so that the reader has a concept of the magnitude of changes that may be expected.

We have also removed all references to 'coarse' projections and focussed on an if/then approach as suggested. We feel this has greatly improved the manuscript. In particular, we focus on presenting a range of possible outcomes and discuss which outcome is the most likely given the evidence. We also discuss how future changes in the sizes and distribution of cattle and other livestock may modify the size of this feedback. We have also added a reference to other ruminants, sheep and buffalo, as there is emerging evidence that they exhibit similar enteric methane responses to reductions in forage nutritive value as cattle. Finally, we have added a detailed limitations section where we discuss the limitations of our work as well as the opportunities for future research which will improve the accuracy of our projections.

**Responses to Reviewer 1**

General comments: The authors present a meta-analysis of forage studies in order to ascertain any impacts of growing conditions on methane production by livestock. Overall I think this study is a valuable contribution by highlighting a positive feedback between temperature increases and methane production by livestock. The clear and succinct project raises many important questions for global methane contributions and feedbacks under future climate scenarios, but could be improved by some additional considerations and clarifications.

Response: We thank reviewer 1 for these positive comments and, in particular, that we have raised some important questions regarding global methane contributions and feedbacks.

Changes made: No changes requested by reviewer.

I did not find that the analysis of Nitrogen addition to add much value as part of the results section, and because of the limited data on this part of the study (and lack of a significant impact for the species with the most data) is too preliminary an analysis for inclusion.

Changes made: We have reduced the emphasis on nitrogen in our results and discussion. Since nitrogen fertilisers are not available to all farmers, we have also now not included nitrogen as a part of the modelling exercise. However, since information on nitrogen addition rates was present for a large proportion of the dataset (67%) and these were shown to be significant in our analysis, we have retained, a short section of results and discussion points relating to the effects of nitrogen fertilisation. We discuss the reasons why N addition may have a significant effect across all species but not when considering just one. We also highlight the size of the nitrogen dataset to prevent future misunderstanding.

Cross reference: 116, 232-234, 238, 341-349.

In addition, clarification and discussion if these results are indicative of changes within a species or between species (and relative contributions of each) is needed as this is considerable factor in the assumptions for their model.

Response: There is substantial variation between species responses to climate and nitrogen, as supported by data that we have presented. However, we were not focussed on these differences, preferring to focus on the effects of climate and fertiliser addition. We therefore included species and site as random effects in our mixed effects statistical modelling. By doing this we accounted for differences between species and between sites without making them the focus of our analysis. We also ran our analysis for the best represented species in the database, *Lolium perenne*. This allowed us to discuss the effects of physiology and phenology separately from species identity. The positive responses to both monthly temperatures and MAT imply that both compositional and physiological changes play a role in determining the observed response. We feel that our models are robust since they are based on an empirical model which includes species turnover, and well as changes to physiology and phenology.

Changes made: We have added more detail in the manuscript so that these methodological aspects are clearer. We have also discussed our results in light of this limitation.

Cross reference: 131-133, 136-142, 150, 234-238, 330-339.

Furthermore, projections related to RCP 2.6 and 8.5 and increased methane should be developed further than currently presented to clarify the relationship to livestock assumptions in these models and present the spatial variability between regions of the world in more detail.

Response: We feel that it is pragmatic not to add more detail on the spatial variability of our projections because of the concerns of both the editor and Reviewer 2.

Changes made: We have added considerable additional detail on the assumptions and limitations of our approach. We have also re-run our analyses using multiple published models, as detailed above. We have added some discussion relating to geographical variation. We also highlight the differences in our projections based on RCP 2.6 and RCP 8.5.

Cross reference: 155-181, 185-188, 276-282, 377-384.

Finally, additional clarification of other assumptions and limitations to this study are needed to generate discussion and thoughts about taking these coarse projections further. Grassland communities are complicated and although the authors show a response to general long term temperature to forage nutritive value, interannual and geographic variability (plus management) are additional important factors.

Response: We agree with this comment.

Changes made: We have included more detail about the assumptions and limitations of our work, including the points made regarding inter-annual and geographic variability and management.

Cross reference: 401-416.

More detailed comments on specifics sections of the manuscript follow. Specific Comments Line 42, if 48% of the biomass is grass, would be good to know what composes the other 52%. This is a big deal for methane production and would help with conclusions and discussion points.

Response: We have additional detail: "Pasture contributes 48% (2.3 billion tons) of the biomass consumed by livestock, followed by grains (1.3 billion tons, 28%). The remainder is from leaves and stalks of field crops, such as corn (maize), sorghum or soybean (Herrero et al., 2013)". We have also discussed how the trend towards feeding mixed diets, which contain a mixture of components, may also further increase methane production.

Changes made: Manuscript text updated to include this information.

Cross reference: 42-44, 374-376.

Line 51, consider talking about tundra regions here as well since you base your results on this climate type. Since these are harsh climate do they behave like arid regions (stressful) or temperate regions (cooler, so greater nutritive value)?

Response: The dataset included data from tundra zones (35 % of entries) and therefore these have informed the relationships we identified. We have also highlighted that data collected from tundra sites are consistent with the relationships generated across all sites, and we also highlight data gathered from the lowest and highest temperatures which additionally support our conclusions.

Changes made: Reference to tundra has now been included in the manuscript to make it clear that the dataset includes this biome.

Cross reference: 57-59, 114, 226-230.

Line 84, you need to talk about the size of the database here and not just percentages. It is important to know the distribution and number of species across climate types, the amount of data that your fertilizer model is based off of, etc. It is hard to determine if the results you have are from within species variability or across species variability. The two lead to different conclusions and are an important to discussing changes in methane production from cattle in the same locations (are we assuming a change in forage species?).

Response: We agree that this is essential for the reader and apologise for the lack of information presented in the manuscript.

Changes made: We have provided a great deal of further detail of how these data break down across biogeographic regions and species in the revised manuscript.

Cross reference: 110-117, 136-142, 234-238.

Line 91, a brief discussion of whether harvested time impacts DM and other variables and then later on, account for this in the analyses (i.e. on line 109 it is reported that a sample was taken at -5 degrees C).

Changes made: We now briefly discuss harvest time in the introduction and discussion. We also highlight values that will have been sampled in very low and very high temperatures.

Cross reference: 59-60, 325.

Line 143, is this for all temperature and rainfall values? Both the month of collection and mean annual values?

Response: Separate analyses were carried out for monthly and annual values.

Changes made: We have added detail to clarify this. We also now include a comparison of the results of monthly and annual temperatures and how they impact on size of the presented relationships. Since they are of similar magnitude we feel this adds confidence to our conclusions.

Cross reference: 130-133, 136-142, 150, 226-238, 330-339.

Line 167, RCP 2.6 and RCP 8.5 incorporate projections in the amount of livestock as a part of determining changes to radiative forcing. Be explicit here that you are restraining your analysis to just the projected temperature changes as determined by RCP 2.6 and then 8.5, not any changes related to projections in number of livestock in the scenarios or any assumptions about where, feed type, etc.

Changes made: The manuscript has been updated to make this point clear. We have discussed how increasing the livestock inventory may additional increase methane production to a greater extent than we project. We also highlight that the future distribution of livestock is unknown, but the location of increases in cattle numbers are important. We have also mentioned that changes in feed composition will also changes methane production.

Cross reference: 24, 281, 401-416, 420, figure 5 caption.

Line 201, consider splitting the relationship between C3 and C4 plants here as you do in the model later on. Looks like a different response but hard to tell.

Response: We have split this in the figure. We do demonstrate in the results that C3 and C4 plants differ, which is an interesting finding from our study.

Cross reference: 313, figure 1.

Line 203, please revise the table caption to better reflect the four models presented. The comparison to the results section and why the numbers of sites differ, plus the two models for NDF and CP are not clear.

Changes made: The table caption and the text have been modified to address this concern.

Line 223, please clarify the figure explanation, it is hard to determine where the two scenarios come from in your temperature model for each size of livestock. Also consider some clarification in the methods section where you present equations for these (line 150).

Changes made: We have removed this figure and added two additional figures. Scenarios and methods are now presented more clearly.

Cross reference: 155-181,185-188, figures 3 and 4

Line 223, I find the nitrogen addition discussion distracting and not needed for the main part of this paper. I think you could make a great point focusing on temperature and save discussion of nitrogen addition to the discussion. It complicates the methods section (data collection) and this is a small part of your database (8%), plus you find a temperature impact for the main species in your data, but not a nitrogen effect (making this a more complicated question).

Changes made: We have reduced the emphasis on nitrogen in our results and discussion. Since nitrogen fertilisers are not available to all farmers, we have also removed nitrogen from the modelling exercise. However, since nitrogen addition rates were included in a large proportion of the dataset (67%) and was shown to be significant in our analysis, we have retained, briefly, some of the results and discussion points relating to the effects of nitrogen fertilisation. We also highlight the size of the nitrogen dataset and discuss the differences between our analysis across all plant species and the best represented plant species.

Cross reference: 116, 232-234, 238, 341-349.

Line 235, I like the analyses but the figures presented could be more informative. In this case these figures mainly represent areas with larger projected temperature change. Consider some alternative presentation, such as presenting the % change by continent, or other factor. A table or figure that presents changes by geographic location for different sizes of cattle would give much more information than currently presented in the text and figure. You could even consider ramifications of increased numbers of livestock in addition to the temperature impacts (as referred to in the discussion but not presented in the results).

Response: We have not added more detail on the spatial variability of our projections because of the concerns of both the editor and Reviewer 2.

Changes made: We have re-run our projections using multiple models which represent a much larger geographic coverage than in our original article. Since these models contain data from multiple regions then we consider it inadvisable to make geographically explicit predictions. We have discussed the ramifications of increased numbers of livestock in more detail in the discussion. We have estimated the possible increase in methane production from a larger cattle inventory and discuss how this value may be modified by the relationships we have identified.

Cross reference: 276-282, 377-383, 409-414, 420-427.

Line 253, talk here a bit more about the assumptions in the model you have created (data sources, species variability vs community variability, forage type, etc.). Again, I think this is a valuable study and addition, just need to explain what additional information is needed to go beyond the "coarse projections."

Response: We agree with this comment and thank the reviewer for their positive comments.

Changes made: We have included more information on our assumptions in the methods and also provide a discussion of these issues.

Cross reference: 23-25, 155-181, 401-416, 423.

**Line 300, what is the magnitude difference of increased methane in housed cattle vs. the increase of methane from grass at warmer temperatures. Can you say your overall projection may increase?**

Changes made: We have limited our analysis to climate driven changes to forage quality and methane emissions, however, the trend towards housed cattle fed mixed diets will likely further increase methane production to a greater extent than we predict. We have presented the magnitude of this difference in the revised manuscript. "Our calculations are limited to cattle which consume grass, however, the trend towards permanently housed cattle, particularly across Europe and North America, may further increase these values because the mixed diets of housed cattle increase enteric methane production by around 58 %".

Cross reference: 374-376.

Line 331, I liked the discussion overall, and think you cover a lot of good points about the conclusions of the study. Two additional factors to consider are the unknowns of the impact of increased CO2 on NDF and CP for grass species (especially C3), how would this impact your conclusions. And secondly, consider a discussion about grazing pressure (which I know you excluded) changing community composition and species response, and those impacts to CD and NDF.

Response: Thank you for your positive and useful comments.

Changes made: We have added discussion of these two crucial points.

Cross reference: 401-416.

**Responses to Reviewer 2**

General comments: The study aims at investigating the relationship between forage quality, methane emissions from livestock, and projected future emissions. The topic is interesting, relevant and timely. The authors have done a good job gathering data to show the variability of forage quality for key quality parameters, plant species and across world climates. And that in itself would be useful material to be published (e.g. Fig. 1 and 2, Table 1) in a specialized forage science journal.

Response: We thank Reviewer 2 for these positive comments. We agree that we have collated a large dataset as summarised above and that the topic is interesting, relevant and timely.

Changes made: No changes requested by reviewer.

What I find less robust, is the use of statistical models derived with forage quality data, and the temperature under which the forage was sampled, to make (future) predictions of methane emissions by livestock. The analyses that would make this manuscript relevant for Biogeosciences are based on a few equations (derived from statistical analyses) which related methane emissions to the quality of the feed. Temperature is an explanatory variable that was used by other authors (Hirotaka Kasuya and Junichi Takahashi – see Asian-Aust. J. Anim. Sci. Vol. 23, No. 5: 563 - 566) to explain the intake of NDF, whereas methane emissions are driven by the intake of NDF. I find the extrapolation of these equations too week to make global predictions of methane emissions.

Changes made: We have presented a detailed summary of our solution to these criticisms above, i.e. we have re-run our predictions using several other published empirical models, including those mentioned in this Reviewers' comment. These additional analyses show the overall finding of the paper are robust. However, we have discussed how these models differ in their predictions, and present a range of possible values for future methane production as a result. We have also reduced our emphasis on global predictions and highlighted the limitations to our approach. We discuss the gaps in current knowledge and call for further research which focusses on closing these knowledge gaps, thus improving future modelling efforts.

Cross reference: 155-181, 290-296, 351-376, 401-416, figures 3 and 4.

This sort of study, interesting and relevant, would be better substantiated using vegetation models that represent the physiological processes through which temperature would affect feed quality, and livestock models that would describe the effect of temperature on livestock (heat stress?) affecting the emission of methane. There are more weaknesses in the assumptions used for the study, which I describe below under specific comments. Unfortunately, I don't find this manuscript suitable for publication in Biogeosciences.

Changes made: we have added a detailed limitations section in which these points are addressed. At present there are no mechanistic models which can be applied to plant nutritive quality or enteric methane production. Empirical models are therefore the best way to make predictions for future forage-driven changes to methane production. We now highlight how several factors, including heat stress, CO2 enrichment, grazing pressure and the frequency of extreme weather events may also influence methane production but further research is needed to quantify the direction and magnitude of these effect. Nevertheless we believe that our analyses and projections identify an important and interesting relationship between climate, forage nutritive quality and methane production.

Cross reference: 23-25, 155-181, 185-188, 401-416, 421-423.

Specific comments L108: the authors used temperature at time of sampling, mean annual temperature (MAT) and monthly rainfall (MAR) over the past 10 years. The quality of the forage is associate to the current growing

season, most like a seasonal and cumulative effect. So the use of an average long term (10 years) temperature of the temperature of the month of sampling seem inappropriate as predictors of feed quality.

Response: The use of MAT in this context allows us to link our statistical models to future climate models, which present predictions in terms of MAT. We have now added a comparison of the results from MAT and sampling temperature in the results section. Sampling temperature is a useful proxy for conditions at the time of sampling when compared with MAT which represents prevailing climatic conditions. We show how MAT and sampling temperatures are both significantly related to forage quality. We also demonstrate that MAT and sampling temperatures resulted in similar outputs and this gave us additional confidence that our conclusions are robust. We have also added detail of these comparisons to the discussion.

Cross reference: 131-133, 136-142, 150, 234-238, 330-339.

L143-148: the use of equations developed for one experiment conducted in Japan, with a limited set of feedstuff (only 4 temperate climate species) to extrapolate global methane emissions seem largely inadequate for the purpose.

Response: This comment has been addressed previously. We have now calculated our predictions using an expanded suite of models. While predictions vary between these models, the overall picture is one of support for our original conclusion.

Changes made: As above.

Cross reference: 155-181, 290-296, 351-376, 401-416, figures 3 and 4.

L192: I would have expected a species effect in the analyses of NDF. Under the same climate and soil there will be plants with largely different values of NDF, and other quality parameters simply because of genetic differences.

Response: There is substantial variation between species responses to climate and nitrogen, as supported by data that we have presented. However, we were not focussed on these differences, preferring to focus on the effects of climate and fertiliser addition. We therefore included species and site as random effects in our mixed effects statistical modelling. By doing this we accounted for differences between species and between sites without making them the focus of our analysis. We also ran our analysis for the best represented species in the database, *Lolium perenne*. This allowed us to discuss the effects of physiology and phenology separately from species identity. The positive responses to both monthly temperatures and MAT imply that both compositional and physiological changes play a role in determining the observed response. We feel that our models are robust since they are based on an empirical model which includes species turnover, and well and changes to physiology and phenology.

Changes made: We have added more detail in the manuscript so that these methodological aspects are clearer. We have also discussed our results in light of this limitation.

Cross reference: 131-133, 136-142, 150, 234-238, 330-339.

L232: the use of the selected statistical models derived from one single experiment, with future temperatures seem inappropriate to predict both future and actual methane emissions globally.

Response: This comment has been addressed previously. We have now calculated our predictions using a suite of models.

Changes made: As above.

L247: I disagree with the authors. They don't describe here a climate feedback, but an artefact of the use of statistical models and projected temperatures. The relationship between temperatures and plant quality parameters is largely known in ecology. That explains the differences between ecotypes across the globe. However, the authors extend these relationships to the calculation of methane emissions, and that seems incorrect.

Response: We do not believe this is a statistical artefact. The logic is robust at each stage and the relationships presented are supported by previous published work. If the relationship between temperatures and plant quality is largely known and the relationship between plant quality and methane production well resolved then we believe that the overarching concept we have identified is robust. As summarised in a recent review paper, elevated forage NDF has been consistently demonstrated to have a positive effect on methane production in cattle across many studies (Appuhamy et al., 2016).

Changes made: We have highlighted the consistent link between forage NDF and methane production.

Cross reference: 49, 155-181, 185-188, 312-314, 360.

L264: The differences in NDF and CP across climate doesn't mean that ruminants are under nutritional stress. Livestock keepers manage different species and breed adapted to their climate across the globe. And therefore it is not correct to use one equation derived for Bos taurus dry cows in Japan to predict global emissions of ruminants

Response: We have removed his reference to nutritional stress and commented simply that the nutritive value of forage grasses in warmer regions is lower. We have also discussed that there is a lack of data relating to the effects of heat stress on enteric methane production in our limitations section. Finally, we have now re-run our projections using multiple published models, as stated above. These now cover many different regions and a large number of animals.

Cross reference: 156, 401-416, table 1.

[revised manuscript text omitted]
               | 1       | 5.1 x NDF 2 - 39.3 x NDF + 360.0                                       | -        | А     |
| 2    | D      | EU, NA, AUNZ     | 21      | -2.8 + 3.7 x NDFi                                                                 | 18.3     | В     |
| 3    | D,B    | NA               | 172     | 1.6 + 0.04 x MEi + 1.5 x NDFi                                                     | 17.9     | С     |
| 4    | D,B    | NA               | 62      | 0.2 + 0.04 x GEi + 0.1 x NDF - 0.3 x EE                                           | 17.9     | D     |
| 5    | D      | EU               | 12      | 1.2 x DMI - 1.5 x FA + 0.1 x NDF                                                  | 16.9     | Е     |
| 6    | D,B    | AF, AS, AUNZ, SA | 35      | $-1.0 + 0.3 \times DMI + 0.04 \times DMI^2 + 2.4 \times NDFi - 0.3 \times NDFi^2$ | 31.4     | F     |

\* 1. Kasuya and Takahashi, 2010, 2. Storlien et al., 2014, 3. Ellis et al., 2007, 4. Moraes et al., 2014, 5. Nielsen et al., 2013,

6. Patra, 2015

690 \*\* NDF = neutral detergent fibre (%DM), NDFi = neutral detergent fibre intake (kg day-1), MEi = metabolisable energy intake (MJ day-1), GEi = gross energy intake (MJ day-1), EE = dietary ether extract (%DM), DMI = dry matter intake (kg day-1) and FA = dietary fatty acid (%DM)

\*\*\* As presented by Appuhamy et al (2016) except ref 4 and 6 which were presented within the referenced article

Figure 1: Boxplots of (a) the neutral detergent fibre (NDF) and (b) the crude protein (CP) content of grasses located in bioclimatic zones as described by the Köppen-Geiger Climate Classification system. Significant differences between zones, as identified by LME models, are denoted by different letters (P

---

## Author Response (AR2)

Dear Professor Stoy,

We would like to re-submit a revised version of our article entitled "Forage quality declines with rising temperatures, with implications for livestock production and methane emissions" to your journal for review following a "publish subject to minor revisions" decision. We would like to thank both Reviewers and yourself for taking the time to read our article again and provide us with additional constructive comments. These have significantly improved the manuscript, which we now hope is suitable for publication.

In our revision, we have incorporated almost all of editorial changes that were suggested by the Reviewers. Most notably, we have toned down our claims, particularly in the abstract and discussion sections. We have also clarified the use of Representative Concentration Pathways (RCPs) in our projections. RCPs already include simulated changes in the global cattle inventory (IPCC, 2014). Our results therefore demonstrate that forage and temperature-driven increases in methane production may offset some of the methane reductions assumed to come from fewer cattle (RCP 2.6) or may further amplify methane increases from a greater number of cattle (RCP 8.5). We have also truncated text relating to forage quality and added standard error values as estimates of uncertainty for our results.

We have also now included a reference to support our approach in generating a mean-weighted model. One of the Reviewers felt that some of the new models should not be included in the manuscript because we used constant values, whilst in contrast the other Reviewer felt that the level of understanding and descriptions of sources of uncertainty of the potential methane temperature feedback from cattle is improved by the inclusion of multiple models. We agree with the latter Reviewer and have therefore decided to retain all of the models whilst still relying on the mean-weighted model as the focus for our future methane production estimates. We feel that the use of constant values for these new parameters is appropriate in this case, because their low absolute values or gradients mean that their influence on model outputs is relatively low compared with NDF. This justification is now stated in the text.

In response to the comments we have also added a layer to figure 5 so that it excludes regions predicted to exceed 30 °C for both RCP 2.6 and RCP 8.5, since greater temperatures are outside the range of values included in our dataset. This layer largely overlaps with regions currently unsuitable for livestock and therefore we have retained our estimates of future increases in methane production. We feel this is essential to give the reader an idea of the potential magnitude of any changes that may be anticipated so they can be placed in the context of other global environmental changes. However, we have added a caveat explaining that it may be the case that some regions fall outside the range of our models under greater warming, thus increasing the uncertainty of these estimates

Please see our responses to all Reviewers' comments presented below. Original comments are in black and our responses are in blue.

Yours sincerely,

mark lee

Dr Mark Lee (on behalf of all authors)

**Editor comments**

Decision: Publish subject to minor revisions (17 Feb 2017)

Both referees felt that the revised manuscript represents a substantial improvement and both recommend (in some cases substantial) additional edits to further improve the manuscript. Please revise the manuscript accordingly and outline any changes in a brief letter with an eye toward accounting for the more critical comments of the referees.

**Responses to Reviewer 1**

The authors have addressed most of the comments by the reviewers. However, some weaknesses remain unresolved. I am not convinced the authors present compelling evidence for a new climate change reinforcing feedback, because of the nature of the data, and the sort of analysis that the data allowed. I do see however opportunities for future improvements in this field, and therefore I would recommend the authors revise the paper, tune down their claims a bit and resubmit the manuscript to be considered for publication.

Abstract

L17 Not clear what the authors call 'species turnover'

L17 - Adjusted text to "changes to species identity".

L18 models were used to 'estimate' not to project

L18 - Adjusted text to "estimate".

L19 the analyses 'suggest', not revealed

L19 - Adjusted text to "suggested".

L20 …and correspondingly 'may' increase methane production

L20 - We have added "may".

L26 …[the suggested changes] 'may' reduce this additional source of …

L26 - We have added "may".

Introduction

L81-82 Please re-phrase: I don't understand here the use of 'where possible' "Published statistical models were used to explore the effect of increased temperatures due to climate change to grass nutritive value and cattle methane production"

L82 - We have removed "where possible" as suggested.

Methods

L110. This database only includes 1 dataset from the tropics (Brazil). However, the analysis of methane prediction is done at the global scale (Fig 5)

L192-194, 330-333, 422 - We accept this as a limitation and we have referenced the need for additional data from tropical and equatorial regions in the limitations section. However, the strength of the relationships that we have observed, coupled with evidence from previous mechanistic and empirical studies (e.g. Barrett et al., 2005; Gardarin et al., 2014; Hirata, 1999; Kering et al., 2011; Kipling et al., 2016), suggest that the relationship between declining forage nutritive value and increasing enteric methane production is robust across a wide range of conditions. It should be noted that the additional models we include were themselves meta-analyses and, as such, are based on relatively large datasets. For example, Patra (2015) included 142 mean enteric methane values collected from 830 cattle in 35 studies across Australia, Brazil, India and Zimbabwe (Table 1). In this study, the author only included studies which they describe as from tropical regions.

L175. The use of assumed constant nutritive values reduces the validity of models C, D and E (Table 1) to estimate future methane production. I wonder whether these models should have been excluded from the 'global' analysis

L182 - We feel that it is reasonable to include constant values within models C, D and E. These additional parameters; GEI, MEI, EE and FA, have a relatively small impact on the output of the equations because their gradients (GEI, MEI) or absolute values (EE, FA) are considerably lower than those of NDF. Our projections are based on the mean-weighted model which takes into account projections from all of the models.

L232-236, were the ranges (Fig.2 up to 30 degrees) used in fitting models for NDF respected when making projections using future temperatures? It is not clear from Figures 3 and 4 whether there were ceilings to the temperature change.

L199-200 & figure 5 - Thank you for this recommendation. We have redrawn our map excluding regions which are expected to exceed 30 °C since these regions are outside the range of our dataset. We have described this in the Methods section. Most of this area overlaps the area already excluded as unsuitable for ruminant livestock.

L251-260. I wonder how useful it is to use models with more feed quality variables, when only NDF changes with temperature.

Please see our earlier response to a similar comment above. Note that we included a broader range of projections in response to previous comments to explore the robustness of our results to a wider range of published models, but many of these models included additional parameters.

L275-281. These 'projections' are the weakest part of the manuscript, given all the assumptions and unbalanced nature of the datasets used in the statistical models, with the tropics and Asia heavily under-represented.

L421-427 - We accept this as a limitation and we have referenced the lack of representation of tropical and equatorial regions in the limitations section. We have retained a very limited number of projections so that the reader has an understanding of the order of magnitude of methane increases that are possible so that these relationships can be put in the context of all global environmental changes. We also acknowledge that our findings are preliminary. We have included a detailed limitations section and numerous caveats to our projections.

L330. What do the authors mean with turnover in species identity? I suspect the authors refer to a change in the composition of the plant community drive by increase in temperature – a sort of adaptation mechanism?

L342 - This has been clarified and now says "species compositional (i.e. turnover in species identity), phenological and physiological changes each play a role".

L338. I am not convinced the predictions are robust. The authors find a statistical significant relationship, not a causal relationship.

L50-51, 330-333 - It was not our intention to identify a causal relationship, which requires controlled experimental conditions. This is something which would require significant effort to perform at a global scale. However, at local scales, links between higher temperatures and declining nutritive values, and between declining nutritive values and increasing enteric methane production have been established under controlled conditions (as summarised by Knapp et al., 2014). Our results indicate that the same mechanisms may operate at a global scale. We have clarified this in the text.

L349. I suggest to replace Projections by Explorations

L361 - We have changed the sub-heading to "Explorations of future methane production" as suggested.

L358. Please make sure that the 5 degrees' increase doesn't place the 'predictions' outside the range of validity of the statistical models.

L204, 369 & Figure 5 – We have now stated this in the Methods section. We have also redrawn our map excluding regions which are expected to exceed 30 °C since these regions are outside the range of our dataset. We also now add a caveat stating that "it may be the case that some areas may fall outside the range of our models under greater warming (i.e. those with MAT > 30 °C) increasing the uncertainty of these estimates".

I wonder whether the authors have seen the paper by Caro et al 2016 Mitigation of enteric methane emissions from global livestock systems through nutrition strategies Climatic Change August 2016, Volume 137, Issue 3, pp 467–480 – which is similar to their study

L359 – Thank you for this suggestion. We have made reference to this article in the manuscript "Improved nutrition management by farmers may also partially offset predicted gains in enteric methane production (Caro et al., 2016)."

L403. It seems the authors don't know the literature well. The mechanistic cattle model (RUMINANT) by Herrero et al 2013 (paper cited by the authors) simulates methane emissions for individual animals, of different breeds and for different regions if feeds are known. The LINGRA model maintained by the Plant Production Systems group of Wageningen University (see http://models.pps.wur.nl/node/958) simulates grass quality, and is driven by temperature, rainfall and radiation. It is likely that the combination of both mechanistic models could produce more robust estimates than the ones presented here by the authors.

L421-424 - We are aware of these mechanistic models and also the limitations that are associated with them. For example, they require large amounts of data to parametrise, many of which are unavailable. We have drawn attention to these issues in our discussion, particularly in the limitations section, which we have now modified to read "Current livestock models require many inputs, which are not universally available, and do not account for variation across all individuals, breeds and regions. Furthermore, current mechanistic vegetation models do not quantitatively consider climate-driven changes to forage nutritive quality (Kipling et al., 2013)". Please also note that the LINGRA model is parametrised using data collected exclusively from Europe, predominantly from the UK and only from one species; *Lolium perenne*. The RUMINANT model includes changes to the livestock inventory and several other socio-economic parameters which are outside the scope of our article and are associated with great uncertainty. We remain confident that our results are robust and that our approach is the best that is currently possible considering the availability of data and uncertainty associated with these other models.

L419 Conclusions should be tuned down in ambition. Preliminary evidence is presented here

We have toned down our Abstract, Discussion and Conclusion, making it clear that the results are preliminary.

**Responses to Reviewer 2**

Thanks for the opportunity to take a look at the revised version of this paper. I feel that the revisions have greatly clarified and improved this manuscript. The level of understanding (and sources of uncertainty) of the potential methane temperature feedback from cattle is improved by the inclusion of multiple models. In addition the manuscript covers the implications and limitations of the study more clearly without going beyond their findings. I have made some suggestions for clarity and to explicitly cover assumptions from the RCPs used in this study.

L93 – Do you include data from sites that are in non-cattle producing regions (such as those you exclude in figure 5)? Would be good to clarify this point whether it was an a priori decision (in methods) or just where the data fell out (in the results).

L119 - We have now added "Our dataset represented nutritive values collected from forage plants growing in regions currently suitable for ruminant livestock."

L188 – Why do you choose this combination? Do you have a reference or explanation for why this is a valid measure to weight these models?

L192 - This is a standard approach, weighted according to the number of contributing datasets. We have now added a reference to support this approach.

L191 – This section needs some minor clarification on the use of RCPs. RCP 2.6 includes a reduction in livestock as part of the way to achieve targets. You are keeping livestock constant in your modeled projections. This seems fine if stated, since you use the RCPs to represent temperature increases and relate to current livestock, but the assumption should be stated (and implications discussed). Likewise, RCP 8.5 includes an increase in cattle as part of the inputs into climate models.

L199-200, 290, 369-372 & Figure 5 - Thank you for this clarification. We have now included this to the Methods and Discussion sections, and adjusted our figure.

L200 – 213 – This section can be shortened. The paper is about overall trends independent of species so keep the species specific detail to the appendix (with the exception being Lolium perenne which you model). Maybe just cover the general patterns of NDF and CP from the data which relate to your discussion points (I could see some expansion of this in the appendix to better describe the patterns in NDF and CD).

L210-217 - This section has been shortened so that it only describes the general patterns. We have included considerable detail in the appendix.

L230 – Include some description of model fit and/or confidence intervals

L235-245 - Standard errors of model outputs are now included in the text as measures of uncertainty.

L281 – See comment on line 191. This should say "do not represent changes to the global cattle industry as projected in the RCPs."

L290 - This has been adjusted as recommended.

L350 – See comment on line 191. It is important to convey what your results mean for the assumptions in the RCPs. If the GHGs from RCP2.6 are calculated using fewer cattle (which encompasses a multitude of forcings that change, including methane production) your results demonstrate that temperature increases may increase the methane production offsetting some of the methane reductions assumed to come from fewer cattle. Likewise in RCP 8.5, which includes increases in cattle, your results show methane may be underrepresented since the scenario includes methane from additional cattle, but not a potential increase per head from higher temperatures and forage quality.

L369-372 - We fully agree. We have now added "Since RCPs include changes in the global cattle inventory (IPCC, 2014), our results demonstrate that temperature increases may increase methane production offsetting some of the methane reductions assumed to come from fewer cattle (RCP 2.6) or further amplifying methane increases from a greater number of cattle (RCP 8.5)."

L700 – See line 230

L732 – Values do not include projected changes to the global cattle inventory (up or down relative to RCP assumptions and your other references).

Figure 4 - We have now added "Values do not include projected changes to the global cattle inventory".

L737 – See earlier comments on line 200. Since your study shows a larger range of CP and NDF then prior compilations (an interesting finding in its own right), exploring the data's distribution could warrant more coverage here.

[revised manuscript text omitted]